

# Crypsis and convergence: integrative taxonomic revision of the *Gehyra australis* group (Squamata: Gekkonidae) from northern Australia

Paul M. Oliver[1,2], Audrey Miranda Prasetya[3], Leonardo G. Tedeschi[3], Jessica Fenker[3], Ryan J. Ellis[4,5], Paul Doughty[4] and Craig Moritz[3]

[1] Environmental Futures Research Institute and School of Environment and Science, Griffith University, South Brisbane, QLD, Australia
[2] Biodiversity and Geosciences Programme, Queensland Museum, Brisbane, QLD, Australia
[3] Division of Ecology and Evolution, Research School of Biology, and Centre for Biodiversity Analysis, Australian National University, Acton, ACT, Australia
[4] Terrestrial Zoology, Western Australian Museum, Welshpool, WA, Australia
[5] Biologic Environmental Survey, East Perth, WA, Australia

Corresponding author
Paul M. Oliver,
p.oliver@griffith.edu.au

## ABSTRACT

For over two decades, assessments of geographic variation in mtDNA and small numbers of nuclear loci have revealed morphologically similar, but genetically divergent, intraspecific lineages in lizards from around the world. Subsequent morphological analyses often find subtle corresponding diagnostic characters to support the distinctiveness of lineages, but occasionally do not. In recent years it has become increasingly possible to survey geographic variation by sequencing thousands of loci, enabling more rigorous assessment of species boundaries across morphologically similar lineages. Here we take this approach, adding new, geographically extensive SNP data to existing mtDNA and exon capture datasets for the *Gehyra australis* and *G. koira* species complexes of gecko from northern Australia. The combination of exon-based phylogenetics with dense spatial sampling of mitochondrial DNA sequencing, SNP-based tests for introgression at lineage boundaries and newly-collected morphological evidence supports the recognition of nine species, six of which are newly described here. Detection of discrete genetic clusters using new SNP data was especially convincing where candidate taxa were continuously sampled across their distributions up to and across geographic boundaries with analyses revealing no admixture. Some species defined herein appear to be truly cryptic, showing little, if any, diagnostic morphological variation. As these SNP-based approaches are progressively applied, and with all due conservatism, we can expect to see a substantial improvement in our ability to delineate and name cryptic species, especially in taxa for which previous approaches have struggled to resolve taxonomic boundaries.

## INTRODUCTION

Integrative taxonomic analyses utilise multiple independent data sources to illuminate patterns of phenotypic and genetic differentiation (*De Queiroz, 2007*; *Padial et al., 2010*). Thus, in addition to the traditional morphological characters upon which taxonomy was founded hundreds of years ago, systematists in recent decades have progressively added a growing battery of genetic approaches to analyse evidence from karyotypes, allozymes, mtDNA sequence data and, increasingly common, loci from the nuclear genome (*Oliver, Keogh & Moritz, 2015*). However, genetic species delimitation methods can incorrectly diagnose distinct populations as 'species', especially where dispersal rates are low or sampling is sparse (*Carstens et al., 2013*; *Sukumaran & Knowles, 2017*; *Leaché et al., 2018*; *Singhal et al., 2018*). In this context there is a need to set a higher bar to avoid over-splitting, one approach being to test for negligible gene flow where geographic boundaries of candidate species abut or overlap (*Singhal et al., 2018*). Contingent on sufficient geographic sampling, this is now feasible with increasing access to data from thousands of putatively unlinked nuclear loci (Single Nucleotide Polymorphisms, or 'SNPs'; *Leaché et al., 2014*; *Leaché & Oaks, 2017*; *Melville et al., 2017*; *Unmack et al., 2017*; *Georges et al., 2018*).

The gekkonid lizards in the genus *Gehyra* from Australia show conservative body form, variable appearance and have been a test case for implementing new methods to document species diversity (*King, 1979*, *1982*; *Sistrom, Donnellan & Hutchinson, 2013*; *Sistrom et al., 2014*; *Ashman et al., 2018*; *Kealley et al., 2018*; *Moritz et al., 2018*; summarised in *Doughty et al., 2018a*). The various genetic techniques have led to increasingly finer resolution of phylogenetic structure, and have recently resulted in major revisions of *Gehyra* species-groups from the Australian arid zone and the Australian Monsoonal Tropics (AMT) such that the number of recognised species has nearly doubled from 22 to 43 (*Hutchinson et al., 2014*; *Doughty et al., 2018a*, *2018b*; *Kealley et al., 2018*).

The *Gehyra australis* group is a lineage of geckos from the AMT, originally defined as having key morphological (medium size and undivided lamellae series) and life history (two eggs) features (*Mitchell, 1965*; *King, 1983a*). This group is recovered as monophyletic group in genus-wide molecular phylogenies for *Gehyra* (*Heinicke et al., 2011*). They are relatively large-bodied (to 95 mm snout-vent length) scansorial geckos that occupy arboreal and rocky habitats. Recent analyses of this group based on mtDNA and exon-capture datasets identified numerous divergent lineages or candidate taxa within the eight species currently recognised (*Noble et al., 2018*; *Oliver et al., 2019*). The major mtDNA lineages were supported by *Oliver et al. (2019)* as being evolutionarily independent when applying statistical delimitation methods to the exon capture data. However, most candidate taxa show low levels of morphological differentiation and non-overlapping distributions with other lineages from the same species complexes (*Noble et al., 2018*; *Oliver et al., 2019*). Furthermore, while mtDNA sampling was geographically extensive in these previous studies, the level of mtDNA divergence between some recognised and candidate taxa was moderate (Tamura–Nei distances lower than 10%), and lower than observed between at least some other species of geckos in the same

biome that show evidence of gene flow between lineages (*Laver, Doughty & Oliver, 2018*; *Moritz et al., 2018*; *Oliver et al., 2019*). The sparse sampling of specimens in the exon capture dataset for each lineage, compared to that for mtDNA, also precluded detailed assessment of introgression at geographic boundaries between candidate taxa.

Here, we focus on two species complexes within the greater *G. australis* group: the *G. australis* and *G. koira* species complexes (*Mitchell, 1965*; *Kealley et al., 2018*; Table 1). Within these two species complexes, there is sufficiently dense spatial sampling to allow for a thorough assessment of morphological and genetic diversity across their distributions in the AMT. In contrast, an assessment of *G. robusta* and *G. borroloola* is not possible at this time owing to fewer specimens available, especially at boundaries between lineages (*Noble et al., 2018*; *Oliver et al., 2019*). The *G. australis* complex ranges widely across the AMT (*Wilson & Swan, 2017*) and currently comprises one recognised species (*Uetz, Freed & Hošek, 2019*), but four candidate species of arboreal/generalist taxa (*Noble et al., 2018*). From morphological and ecological perspectives, the *G. koira* complex consists of mostly large-bodied, saxicoline taxa from the western AMT. The two recognised taxa were originally described as subspecies—*G. koira koira* and *G. koira ipsa* (*Horner, 2005*), but have been regarded to be full species in recent field guides (*Wilson & Swan, 2017*; *Cogger, 2018*). Two additional candidate species have also been identified (*Oliver et al., 2017*, *2019*) (Figs. 1 and 2). Prior genetic analyses also revealed a single lineage (koira 4 of *Oliver et al. (2019)*) genetically nested within the *G. koira* complex that has been consistently assigned to the *G. australis* complex on the basis of its small size, colouration and arboreal ecology. Given its morphological and ecological distinctiveness from other species in the *G. koira* complex, we treat this species separately below.

In this paper we present additional molecular analyses to *Noble et al. (2018)* and *Oliver et al. (2019)* to assess detailed patterns of genetic differentiation. Specifically, we were interested to test for introgression where lineages come into contact or overlap by applying SNP analyses, as recommended by *Singhal et al. (2018)*. To do this, we applied a new statistical approach to identifying discrete genetic lineages as deviations from a spatial model of isolation by distance within lineages (conStruct; *Bradburd, Coop & Ralph, 2018*). For low dispersal organisms, this method could be more appropriate than coalescent delimitation methods that assume random mating within lineages (e.g. BPP; *Yang & Rannala, 2010*). Specifically, when applied to continuously sampled populations with parapatric or overlapping sympatric distributions, this method has the potential to test for discrete structure vs. introgression across lineage boundaries. We also assessed habitat preferences and patterns of morphological variation across candidate taxa within these two species complexes of *Gehyra*, focussing on size, scalation and pattern. On the basis of these results, we recognise nine species within the *G. australis* and *G. koira* complexes, six of which are newly described herein.

## MATERIALS AND METHODS

### Sampling and specimens

Within the *G. australis* and *G. koira* complexes nine candidate lineages were identified by previous analyses based on: (a) a geographically comprehensive mtDNA sampling
including 182 individuals in the *G. australis* complex and 183 individuals of the *G. koira* complex, and (b) more focused nDNA (exon capture) analyses including 17 and 23 individuals from the two complexes, respectively (Fig. 1; Table S1; *Noble et al., 2018*; *Oliver et al., 2019*). The mtDNA lineage assignments were used as a basis to: (a) select samples for inclusion in a SNP-based investigation into patterns of differentiation across geographic ranges and of recent gene flow at areas of contact between lineages, and (b) select genetically typed specimens from which to obtain morphological data. For the SNP analyses, we selected one individual per locality to ensure independence of samples. A small number of additional non-genotyped samples from localities that were taxonomically unambiguous were added into morphological analyses to increase sample sizes for key morphological traits (especially pore number in males) for poorly-sampled taxa (Table S1).

Specimens used in this study were sourced from museum collections across Australia (summarised in abbreviations section below), with many coming from our own field work (CCM field numbers, now housed in appropriate museum collections) with incidental observations on habitat use. The lectotype of *G. australis* at the Natural History Museum, London, UK was also examined. Additional samples for which tail tips only were taken are listed in referred material, and most are currently stored at the ANU. All new material was collected under animal ethics approval from the ANU and collection permits from the relevant authorities in Western Australia, the Northern Territory and Queensland.

## Molecular genetics

In light of the overall morphological similarity between many of the lineages identified by previous genetic analyses, we undertook additional genetic analysis using SNP data generated by Diversity Array Technology (DArT$^{TM}$). This method uses restriction-enzyme mediated genome reduction prior to library construction and parallel sequencing (*Jaccoud et al., 2001*) with Next-Generation-Sequencing platforms to sequence the most informative representations of genomic DNA sampling as an alternative to whole genome sequencing, and has proven valuable for detecting introgression between populations in recent studies (*Melville et al., 2017*; *Unmack et al., 2017*; *Georges et al., 2018*). We sampled across the full geographic range of each lineage, with a focus on areas of potential contact as identified by analyses of mtDNA datasets, including 106 individuals in total (72 from the *G. australis* complex and 34 from the *G. koira* complex; Table S1). The candidate lineage koira 4 (from *Oliver et al., 2019*) was not included in these analyses as it was morphologically and ecologically distinctive from other species of the *G. koira* complex.

Sequences generated were processed using proprietary DArT analytical pipelines, including independent SNP calling across sample replicates to estimate repeatability of genotype calls. This pipeline treats the fatq file by filtering poor quality data using stringent selection criteria, generating multiple sequence as reference for marker calls that are aggregated into clusters using the DART fast clustering algorithm with a Hamming distance. Identical sequences are collapsed and low-quality bases in a singleton tag were eliminated or corrected based on multiple sequences as reference. These corrected

**Table 1 Position and name of diagnostic amino acids in the *ND2* sequences for species of the *Gehyra australis* complex and *G. koira* complex.**

| ++ Codon | G. australis complex | | | | G. koira complex | | | | |
| --- | --- | --- | --- | --- | --- | --- | --- | --- | --- |
| | G. australis | G. arnhemica sp. nov. | G. gemina sp. nov. | G. lauta sp. nov. | G. koira | G. ipsa | G. lapistola sp. nov. | G. calcitectus sp. nov. | G. chimera sp. nov. |
| 10 | | | | Thr* | | | | | |
| 19 | | | Thr* | | | Met* | | | Thr |
| 25 | Leu | | Leu | | | Leu | | | |
| 28 | Val* | | | | | | | | |
| 55 | | | | | | | | Met** | |
| 82 | | Met | Met | | | | | | |
| 127 | Met** | | | | | Val | | | |
| 130 | Thr | | Thr | Thr | | | | | |
| 139 | | | | | | | Thr | | |
| 196 | | | | | | | | Thr* | Thr* |
| 205 | Met | Met | Met | Met | | | | | |
| 226 | | | | | Leu | Leu | Leu | Leu | Leu |
| 241 | | | | | | | Tyr | | |
| 253 | | Ala | Ala* | | Ala** | | | | |
| 265 | | | | | | | Pro | | |
| 268 | | Ser | | | | | | | |
| 274 | | Ala* | | | | | | Ala | |
| 277 | | | | | | | Ala | | |
| 286 | | Ile** | | | | | | | |
| 289 | | Val** | | | | | | | |
| 292 | | | | Thr | | | | | |
| 307 | | | | Leu | | | | | |
| 322 | | | | | Met | | | | |
| 370 | | | | | | | | Phe | |
| 376 | | | Ala** | | | | | | Ala |
| 382 | | | | Met | | | | | |
| 406 | | Met* | | | | | | | |
| 415 | | Leu | | | | | | | |
| 418 | | | | | Thr | Thr | | | Thr |
| 433 | | | | | Thr** | | | | |
| 436 | | | | | Tyr** | | | | |
| 451 | His | His | His | Leu | | | | | |
| 454 | Ser** | | | | | | | | |
| 460 | Ile** | Thr* | Ile* | Ile | | | | | |
| 466 | | | Phe | | Ile** | | | | |
| 469 | Val* | Ala* | | | Ala* | | | | Ile |
| 484 | | | | | | | | | |
| 490 | | Val | Phe* | | Phe** | | | | |
| 499 | Leu | Leu | Leu | Leu | | | | | |

(Continued)

| ++ Codon | G. australis complex | | | | G. koira complex | | | | |
| --- | --- | --- | --- | --- | --- | --- | --- | --- | --- |
| | G. australis | G. arnhemica sp. nov. | G. gemina sp. nov. | G. lauta sp. nov. | G. koira | G. ipsa | G. lapistola sp. nov. | G. calcitectus sp. nov. | G. chimera sp. nov. |
| 502 | | | | | | Thr | | | Thr* |
| 559 | | | | | | Thr | | | Thr |
| 583 | Ser* | Ser | Ser | Ser | | | | | |
| 586 | | | | | | | Met | | |
| 589 | | Asn | Asn* | Asn | | Asn | Asp* | | Asn |
| 592 | Gin | Gin | Gin | Gin | | | | | |
| 595 | | | | | Trp* | | | | |
| 616 | | | | | | Ile | | | |
| 619 | Ile | Ile | | Ile | | | | | Val* |
| 625 | Ile | Ile* | Ile | Ile | | | | | |
| 628 | | | | | Met* | | Met | Met | |
| 631 | Thr | Thr | | Thr | | | | | |
| 649 | | | | Leu | Ala* | | | Ala | |
| 658 | | | | Ser | | | | | |
| 661 | | | | Ala | | | | Met* | |
| 664 | Thr | Thr | | Thr | | | | Thr | |
| 667 | Ser | Ser | Ala | | | | | | |
| 679 | | | | | Met | | | Met | |
| 682 | | | | | Lys | | | Lys | |
| 694 | | | | | Met* | | Ile | Met | |
| 697 | | | | | | | | Ala* | |
| 706 | Thr | | | | | | | Thr | |
| 715 | | | | Pro | | | Pro | | |
| 721 | | | | | | | | Met* | Met** |
| 781 | Leu | | Leu | | | | | | |
| 811 | | | | | Ala** | | | | |
| 814 | | | | | | | | | Ala** |
| 820 | | Thr** | | | | | | | |
| 823 | | | | | | Met | | | |
| 826 | | | | | Met* | | | | |
| 832 | | | | | | | | Met* | |
| 835 | Met* | Thr | Thr | Thr | | | | | |
| 838 | | | | | | | Ile | | |
| 841 | | | | | Thr* | | | Gly | |
| 847 | | | | | | | Thr | | |
| 880 | Val* | | Val* | Val | | | | | |
| 886 | | | | | | Val | | | Val |
| 910 | | | | Ile | Ala** | Ala | Ala | Ala* | |
| 937 | | His | | | | | | | |

| ++ Codon | G. australis complex | | | | G. koira complex | | | | |
|---|---|---|---|---|---|---|---|---|---|
| | G. australis | G. arnhemica sp. nov. | G. gemina sp. nov. | G. lauta sp. nov. | G. koira | G. ipsa | G. lapistola sp. nov. | G. calcitectus sp. nov. | G. chimera sp. nov. |
| 961 | Pro | Pro* | | Pro | His | His* | His | His* | His |
| 964 | | | | Asn | | | | | |
| 967 | | | | Arg | | Pro* | | Arg* | Val* |
| 970 | His | His | His | His | | Gly* | | | |
| 976 | | | | Ala | | | Ala* | | |
| 982 | Met | | | | Ala** | Pro* | Ala | Ala | |
| 994 | | | | | Thr | Thr* | Thr | Thr* | Thr |
| 1021 | Met | | Met | | Ala* | | | | |
| 1030 | | Ala* | Ala* | | | Ala* | | | |
| 1033 | | | | Ile | Ile* | | | | |

Notes:
No asterisk, Amino acids without an asterisk are diagnostic for all specimens.
* 1–2 Individuals within the species do not share the character.
** When majority of the individuals in the species but not all share the character.

sequences are analysed on a secondary proprietary pipeline (DArTsoft14), where SNP markers are identified within each cluster to measure the consistency of allele calls examining primarily average and variance of sequence depth, call rate and average counts for each SNP allele, calculating an index of reproducibility for each locus. This pipeline also includes a BLAST, contrasting the sequences with viral and bacterial sequences at GenBank looking for potential contaminants. The final output consists of two files; the SNP calling, including the presence of nucleotide polymorphisms in restriction fragments, and the SiliciDArT file, representing the presence and absence of restriction fragments in each SNP. More details on SNP genotyping can be seen in *Wells & James (2018)* and *Georges et al. (2018)*.

To ensure the quality of the data, all monomorphic sites were excluded and we filtered by repeatability across technical replicates (>99%) and call rate (<10% missing data), we removed duplicate SNPs in the same fragment using the R package 'dartR' (*Gruber et al., 2018*). To visualize the divergence between samples we also generated a distance-based principal coordinates analysis (PCoA) based on the genetic distance matrix and using 'dartR' (*Gruber et al., 2018*). To address the difficulty of determining discrete population structure with isolation by distance, for each species complex we generated conStruct models with a $K$ between 1–7 (*Bradburd, Coop & Ralph, 2018*), but with a focus on the number of candidate lineages. This analysis is a model-based clustering, similar to ADMIXTURE (*Alexander, Novembre & Lange, 2009*), but considers the spatial covariance of genetic data when discriminating discrete populations.

## Morphological analyses
Measurements and scale counts were recorded from 140 adult specimens (Table S2) using Mitutoyo electronic digital callipers (to the nearest 0.1 mm) and dissecting microscope.

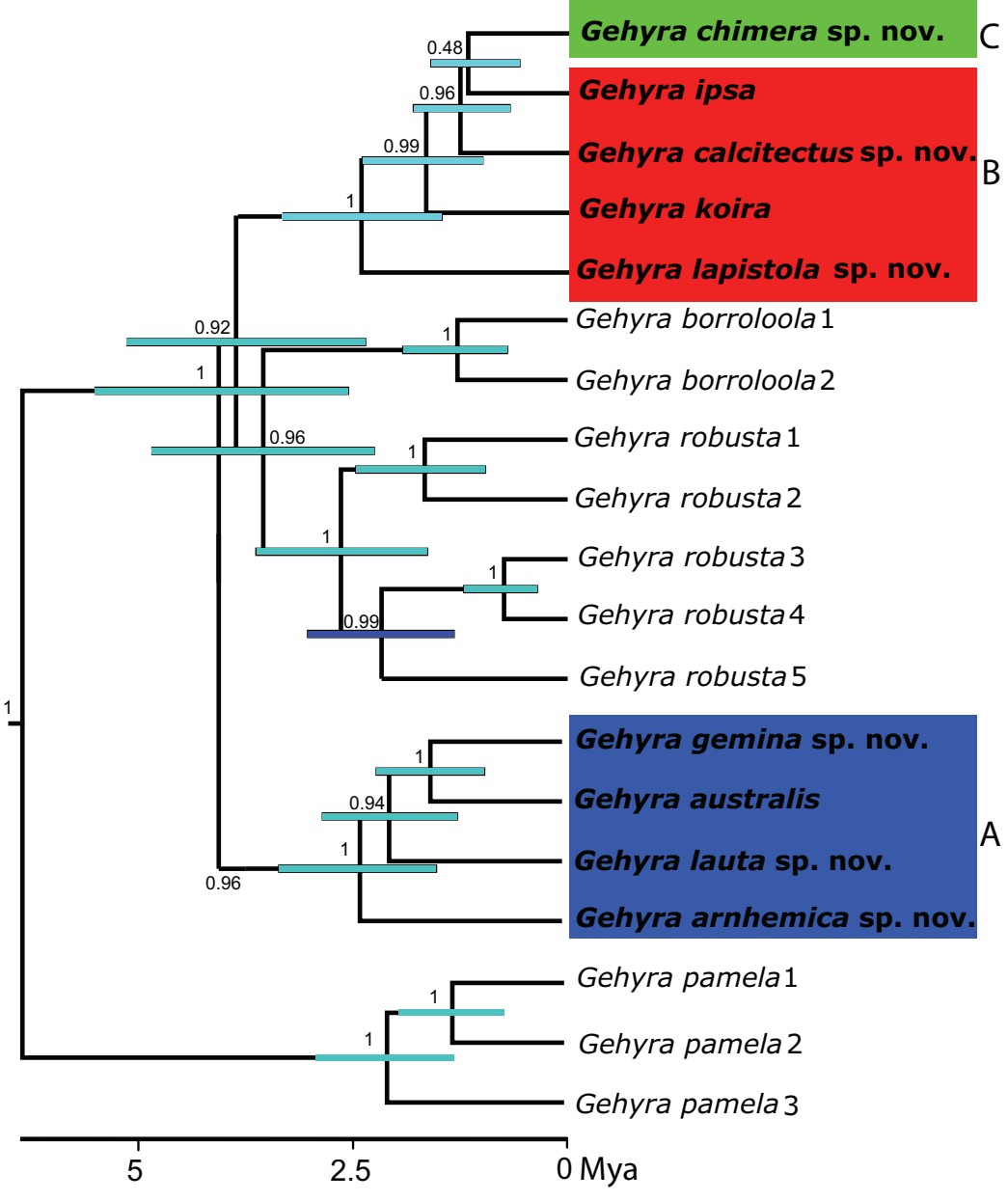

**Figure 1 Phylogeny for the *Gehyra australis* group.** Bayesian coalescent tree (50 exons) from *Oliver et al., 2019*. Highlighted are: (A) *G. australis* complex–blue; (B) saxicoline taxa in *G. koira* complex and (C) *G. chimera* sp. nov.–green.

We measured 11 characters: snout-vent length (SVL), trunk length (TrunkL: body length between forelimbs and hindlimbs), trunk width (TrunkW: width between ventral skin folds of forelimbs), forelimb length (ForelimbL: elbow to base of wrist), hindlimb length (HindlimbL: knee to heel), head length (HeadL: anterior edge of ear to tip of snout), head depth (HeadD: at deepest point posterior to the eyes), head width (HeadW: widest part), snout length (SnoutL: anterior edge of eye to tip of snout, measured at oblique angle), snout depth (SnoutD: deepest part of snout anterior to eyes) and toe length (ToeL:
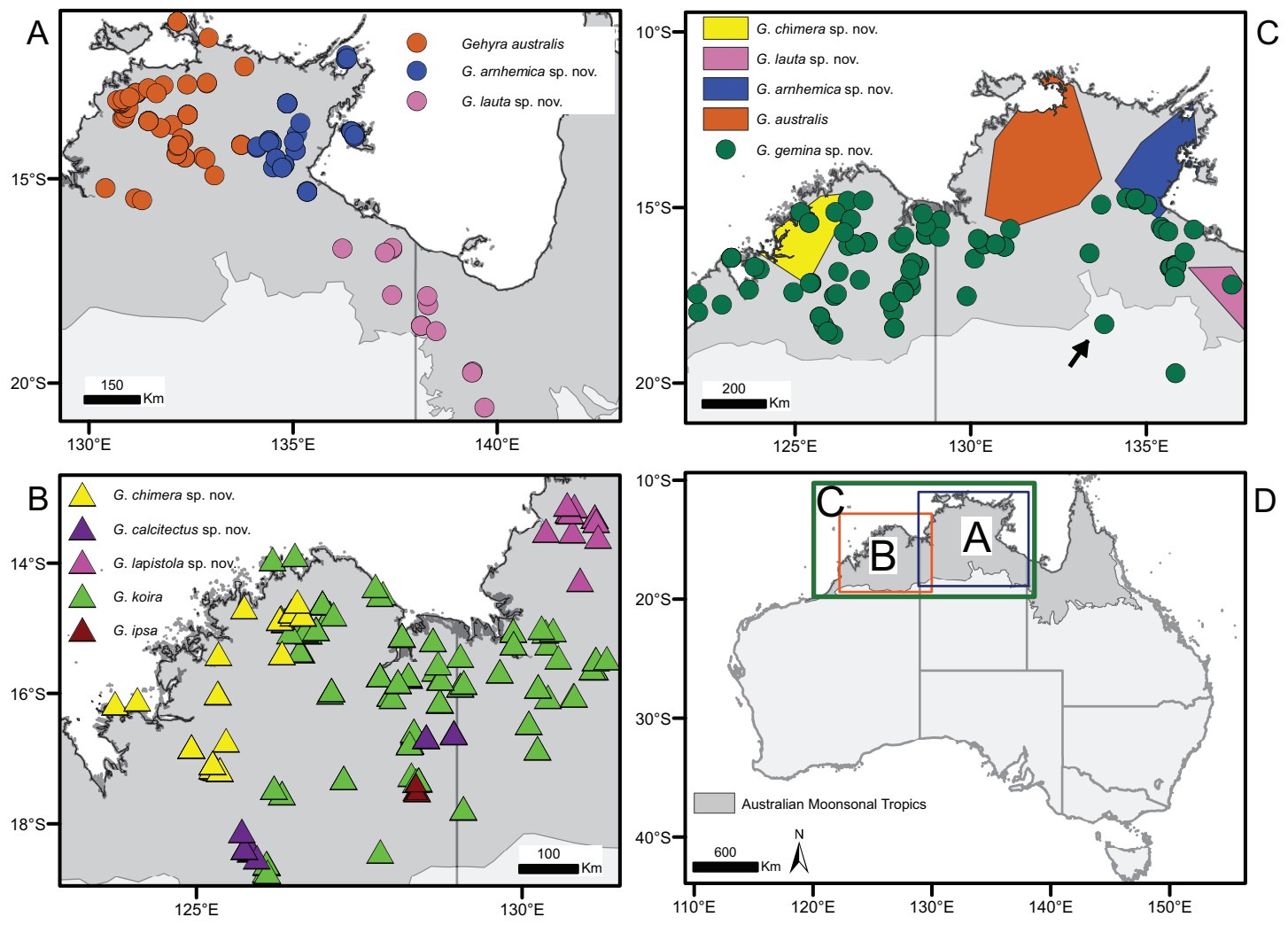

**Figure 2 Distribution of species in the *Gehyra australis* and *G. koira* complexes based on genotyped individuals.** (A) *G. australis* complex taxa from Top End and Gulf regions, (B) *G. koira* complex taxa, (C) *G. australis* complex taxa plus *Gehyra chimera* sp. nov., and (D) map of Australia showing extent of inset regions A–C. Putatively hybrid individual from Renner Springs (NTM R38156; field # CCM1948) is highlighted with an arrow.

base of toe to tip of toe pad on fourth right toe). We also counted number of lamellae on the fourth right toe pad (excluding the apical lamellae), supralabials (to midpoint of eye) and infralabials (to midpoint of eye). Pre-cloacal pores were scored in all male specimens. In this group, as pore number increases, the diameter of distal pores decreases dramatically. Internasals were counted following *Cogger (2018)*, counting all scales between the nares and bordering the top of the rostral. Chin scalation terminology and scoring followed *Hutchinson et al. (2014*; Fig. 4*)*. Relative lengths of first and second chin shields were measured using an ocular micrometer.

Data analyses were done using R (*R Core Team, 2016*) and *R Studio Team (2015)*. All morphometric measurements were log-transformed to improve normality and sexual dimorphism was checked using *t*-tests. No consistent significant indications of sexual dimorphism were found therefore sexes were pooled. Subsequently, transformed values of

the traits were then regressed against SVL to obtain residual scores standardized for body size–these were the values used for further analyses.

Linear Discriminant Analysis (LDA) was then used to iteratively search for distinct combinations of traits that diagnose major lineages. Initial analyses included all species pooled together, with subsequent comparisons focusing on the major complexes (i.e. within the *G. australis* and *G. koira* complexes). The LDAs were done using the MASS package (*Venables & Ripley, 2002*) along with the Caret package (*Kuhn et al., 2012*) for accuracy testing using K-fold cross-validation (0.3 training, 100 repetitions).

Colouration and pattern descriptions were made based on both preserved specimens and photographs of live or recently euthanised specimens where available. Live specimens undergo significant temporal shifts in the intensity of colouration, ranging through pinkish with no colour pattern to darker with obvious patterning over periods of 12 h or less. This colour transition has been reported for many species of *Gehyra* and may be linked to factors such as background substrate and time of day (*Skipwith & Oliver, 2014*; *Oliver et al., 2016b*; *Kealley et al., 2018*). Diagnostic differences in pattern among species were generally apparent when individuals were darkly pigmented. We term this darker pigmented state the 'base colouration'. Unless otherwise reported, descriptions of key aspects of dorsal colouration and patterning tend to focus on animals at the base colouration.

## Rationale for species recognition

We recognise species based on evidence from multiple independent data sources for a history of evolutionary independence (i.e. the generalised lineage concept sensu *De Queiroz, 2007*). We define species as lineages that satisfy two or more of the following criteria: (i) statistically-supported reciprocal monophyly in nDNA phylograms and corresponding support from coalescent delimitation tests (in *Oliver et al. (2019)*, and see below); (ii) evidence from SNPs for discrete population structure especially around parapatric boundaries; or (iii) diagnostic morphological characters based on post-hoc analyses of groups that satisfy (i) or (ii). We note that lack of gene flow, supported by multiple lines of evidence and sufficient sampling of geographic space and genes, is sufficient to delimit species. As the genetic evidence is more compelling than the morphological evidence for several of the *Gehyra* species described here, we provide a table of diagnostic genetic markers to provide character-based definitions of species that are code-compliant (Table 1).

The electronic version of this article in portable document format will represent a published work in accordance with the International Commission on Zoological Nomenclature (ICZN), and hence the new names contained in the electronic version are effectively published under that Code from the electronic edition alone. This published work and the nomenclatural acts it contains have been registered in ZooBank, the online registration system for the ICZN. The ZooBank LSIDs (Life Science Identifiers) can be resolved and the associated information viewed through any standard web browser by appending the LSID to the prefix http://zoobank.org/. The LSID for this publication is: urn:lsid:zoobank.org:pub:9EA86EF0-DB81-40ED-9DB9-58DBEF9B59D6. The online version

of this work is archived and available from the following digital repositories: PeerJ, PubMed Central and CLOCKSS.

To avoid as much repetition as possible in the descriptions and diagnoses of the taxa in this morphologically conserved group, we provide brief diagnoses that apply to all or most member of the two species complexes. In the diagnoses, we provide detailed comparisons of the focal taxon to other similar geographically proximate taxa to facilitate discrimination of species for users. The descriptions are brief, and refer to the diagnoses of the species complex, the species diagnoses and to the tables of genetic and morphological characters. Colouration and pattern are presented in detail for each species, as variation can be subtle and requires more detail to convey differences.

In the following results we use species names as per taxonomic treatments below; these cross reference with the lineage names used by *Oliver et al. (2019)* as follows:

australis 1 = *Gehyra gemina* sp. nov.
australis 2 = *Gehyra australis* Gray, 1845
australis 3 = *Gehyra arnhemica* sp. nov.
australis 4 = *Gehyra lauta* sp. nov.
koira 1 = *Gehyra koira* Horner, 2005
ipsa = *Gehyra ipsa* Horner, 2005
koira 2 = *Gehyra lapistola* sp. nov.
koira 3 = *Gehyra calcitectus* sp. nov.
koira 4 = *Gehyra chimera* sp. nov.

# RESULTS

## Genetic species delimitation

The results of mitochondrial-and exon-based phylogenetic analyses are presented elsewhere (*Noble et al., 2018*; *Oliver et al., 2019*). In brief, the major mtDNA lineages within each species were found to be monophyletic in concatenated nuclear gene phylogenies (with 2–11 individuals per lineage and 1,634 loci) and were supported as separate taxa using coalescent methods (50 loci). There was complete concordance of lineage membership across mtDNA and aggregate nDNA datasets. Accordingly, mtDNA barcodes provide accurate diagnostics for the candidate species. Diagnostic amino acids for each candidate species were identified in the *ND2* gene and are presented in Table 1.

After filtering the DArT SNP data, we obtained a total of 11,113 SNPs for 72 individuals of the *G. australis* complex and 9,516 SNPs for 34 individuals of the *G. koira* complex. In an initial PCoA including all samples across both species complexes, there was strong differentiation among the four lineages in the *G. australis* complex in all PCoA axes, but less differentiation among members of the *G. koira* complex. There was little evidence of recent gene flow between all the eight candidate taxa that were included in the analysis (i.e. no obviously intermediate samples between major clusters), including many samples from areas of parapatry and overlap.

Subsequent analyses focused on differences within each species complex. For the *G. australis* complex, the first principal coordinate of the PCoA accounted for 37.6% and the second for 28.6% of the variance (66.2% in total) (Fig. 3A). Crucially, in samples in

overlap or parapatric zones, there was no sign of admixture, supporting the evolutionary independence of all four candidate species. We observed little evidence of differentiation (i.e. subclusters) within any of the four candidate species in the *G. australis* complex; one exception was slight separation of individuals of *G. gemina* sp. nov. sampled from the Gulf and Barkly Tableland region vs. those from the Victoria River District and westwards. The strongest outlier was a specimen from the northern edge of the arid zone in the NT (NTM R38156 (field # CCM1948)), and far from the known range of the other lineages, that appeared intermediate between *G. gemina* sp. nov. and *G. australis* and had mtDNA from the former (arrow in Fig. 3A). The ConStruct result at *K* = 4, the number of candidate species, presented the same outcome with four distinct lineages. Even in areas where *G. genima* sp. nov. overlap with the other candidate species, there is no admixture, as highlighted in the pie charts, except for the same sample as above (Fig. 3A).

In the *G. koira* complex the first two principal coordinates of PCoA accounted for 41.6% and 20.8% of the variance, representing 62.4% of the variance in the total (Fig. 3B). Each of the four candidate species included forms a highly discrete cluster in this complex with no evidence of any intermediates or outliers. As in the *G. australis* complex, geographically proximal samples from different taxa cluster closely with other samples of the same lineage. Visual inspection, however, suggests genetic heterogeneity within candidate species in the *G. koira* complex, notably across the range of the widespread *G. koira* and between two geographic isolates of the limestone-associated *G. calcitectus* sp. nov. The ConStruct result at *K* = 4 (again, the number of candidate lineages) yielded less consistent results. This analysis largely distinguished *G. koira* from a combination of geographically disjunct populations of the distantly related *G. calcitectus* sp. nov. and *G. lapistola* sp. nov. (Fig. 1), however *G. ipsa* appeared as a mix of the other taxa (Fig. 3B). As ConStruct relies on sufficient geographic sampling to estimate effects of isolation by distance, we speculate that this result may reflect an interaction between the small samples sizes for some taxa (especially *G. ipsa*) and the geographic disjunction within other taxa (especially *G. calcitectus* sp. nov). These two factors may effect the estimation of ancestral admixture within ConStruct (*Bradburd, Coop & Ralph, 2018*).

## Morphological analyses

Multivariate analyses (LDAs) of general body morphology indicated that there was substantial overlap in patterns of morphological variation, except for the large bodied *G. ipsa* (Fig. 4A). One notable feature of the combined analysis is that the tree-dwelling *G. chimera* sp. nov. was distinctive from closely related lineages in the otherwise saxicolous *G. koira* complex, but overlapped more extensively with lineages in the mostly arboreal *G. australis* complex (see also *Oliver et al., 2019*). Overall, there is less separation in body size and shape in the *G. australis* complex than in the *G. koira* complex (Figs. 4B and 4C). In the latter, there is clear separation between individuals of *G. ipsa*, *G. chimera* sp. nov., *G. calcitectus* sp. nov. and *G. lapistola* sp. nov., but *G. koira* overlaps the last two extensively.

Despite this overall lack of differentiation in multivariate analyses, univariate comparisons suggested trait differences among lineages (Fig. 5). In comparisons including

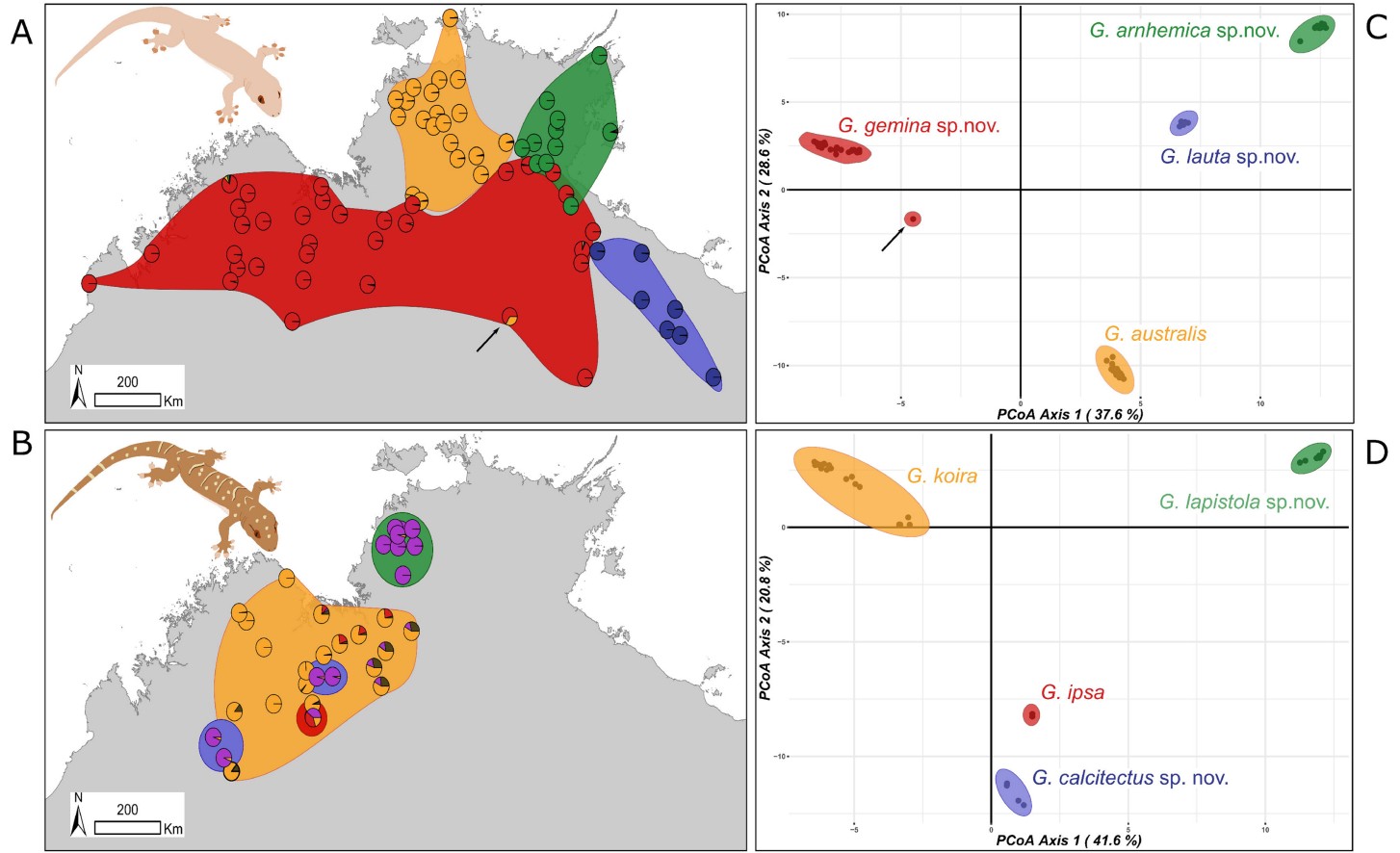

**Figure 3 Sampling and clustering results of DArT analyses.** Pie charts on the map indicate the ConStruct results, with the background polygon colour corresponding to the species groups on the Principal Coordinates Analysis (PCoA) for (A and C) G. australis complex with putative hybrid at Renner Springs indicated with arrow (NTM R38156; field # CCM1948) and (B and D) G. koira complex.

taxa in both of the two species complexes, members of the *G. australis* complex (except for the large-bodied *G. lauta* sp. nov.) were generally smaller than the lineages in the *G. koira* complex, and also tended to have shorter hind limbs (Fig. 5). Within the *G. australis* complex males of *G. arnhemica* sp. nov. and *G. lauta* sp. nov. both had relatively high numbers of pre-cloacal pores in highly tapered series (outer pores much smaller than inner pores) (Fig. 6; Table 2). Within the *G. koira* complex, compared to the nominate taxon *G. koira*, *G. ipsa* was considerably larger (Fig. 4C) and had a lower number of pre-cloacal pores, *G. lapistola* sp. nov. had a lower number of pores, and *G. calcitectus* sp. nov. had both relatively wider head dimensions and lower number of pores (Table 2; Fig. 5).

Examination of scalation also suggested some differences in head and chin scale variation (Fig. 7). Specifically, in the *G. koira* complex, *G. ipsa* often, but not always, had a characteristic enlarged scale between the inner postmentals (see more discussion of this character in the systematics of the *G. koira* complex below).

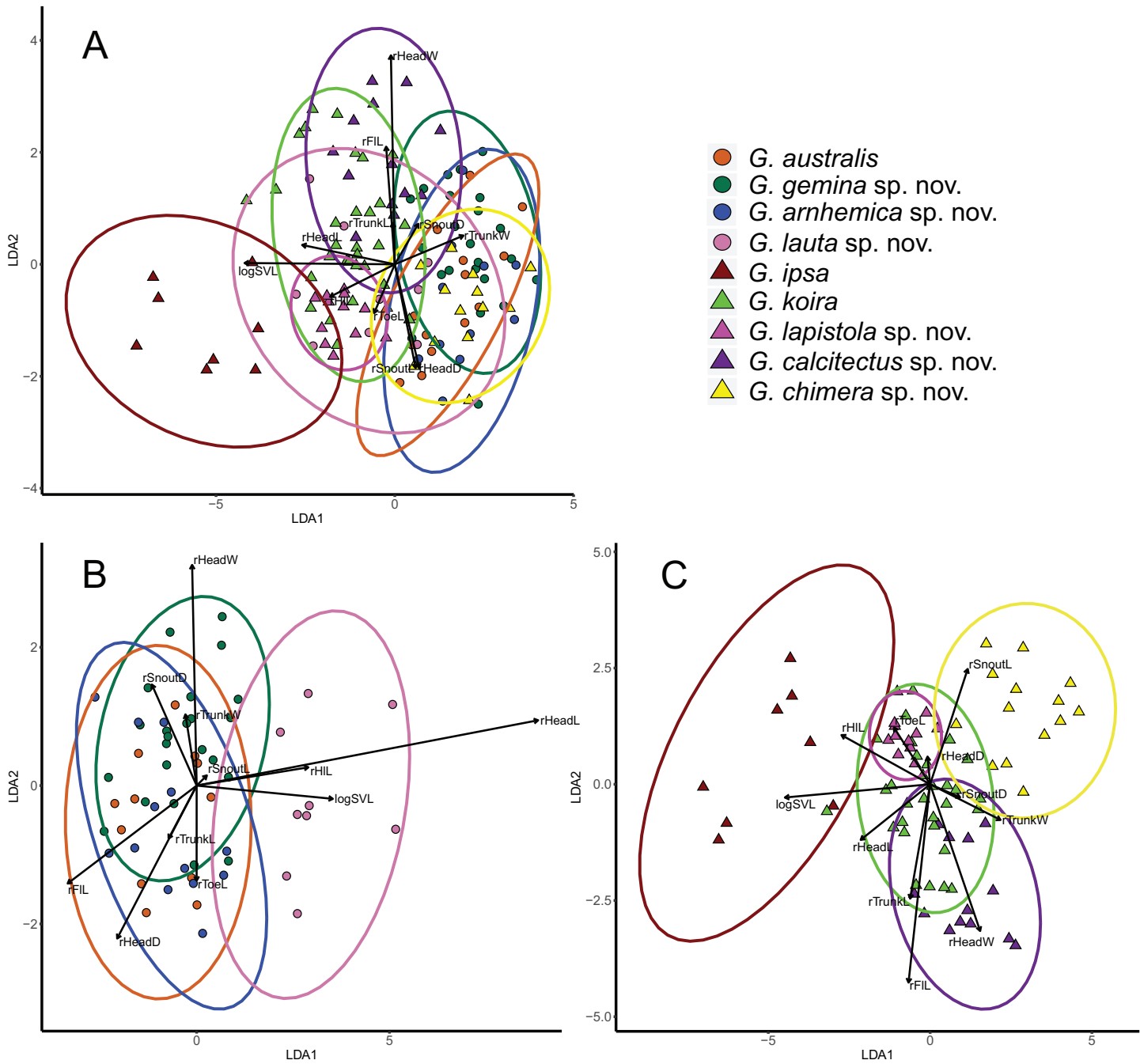

**Figure 4 Visualisation of outcomes of linear discriminant analysis on morphological traits of *Gehyra australis* group specimens.** (A) taxa in the *G. australis* and *G. koira* complexes (koira 4); (B) taxa in the *G. australis* complex; and (C) taxa in the *Gehyra koira* complex including the arboreal *Gehyra chimera* sp. nov. (koira 4). Vectors indicate the relative contributions of strongly loading traits for the first 2 LDA axes.

In colour and pattern there were again some differences, generally more apparent in the base colouration in life than in preservative (Figs. 8–10). In the *G. australis* complex, *G. gemina* sp. nov. and *G. arnhemica* sp. nov. frequently had a dorsal pattern of blotches, barring or vermiculations, while adult *G. lauta* sp. nov. were plain and unpatterned

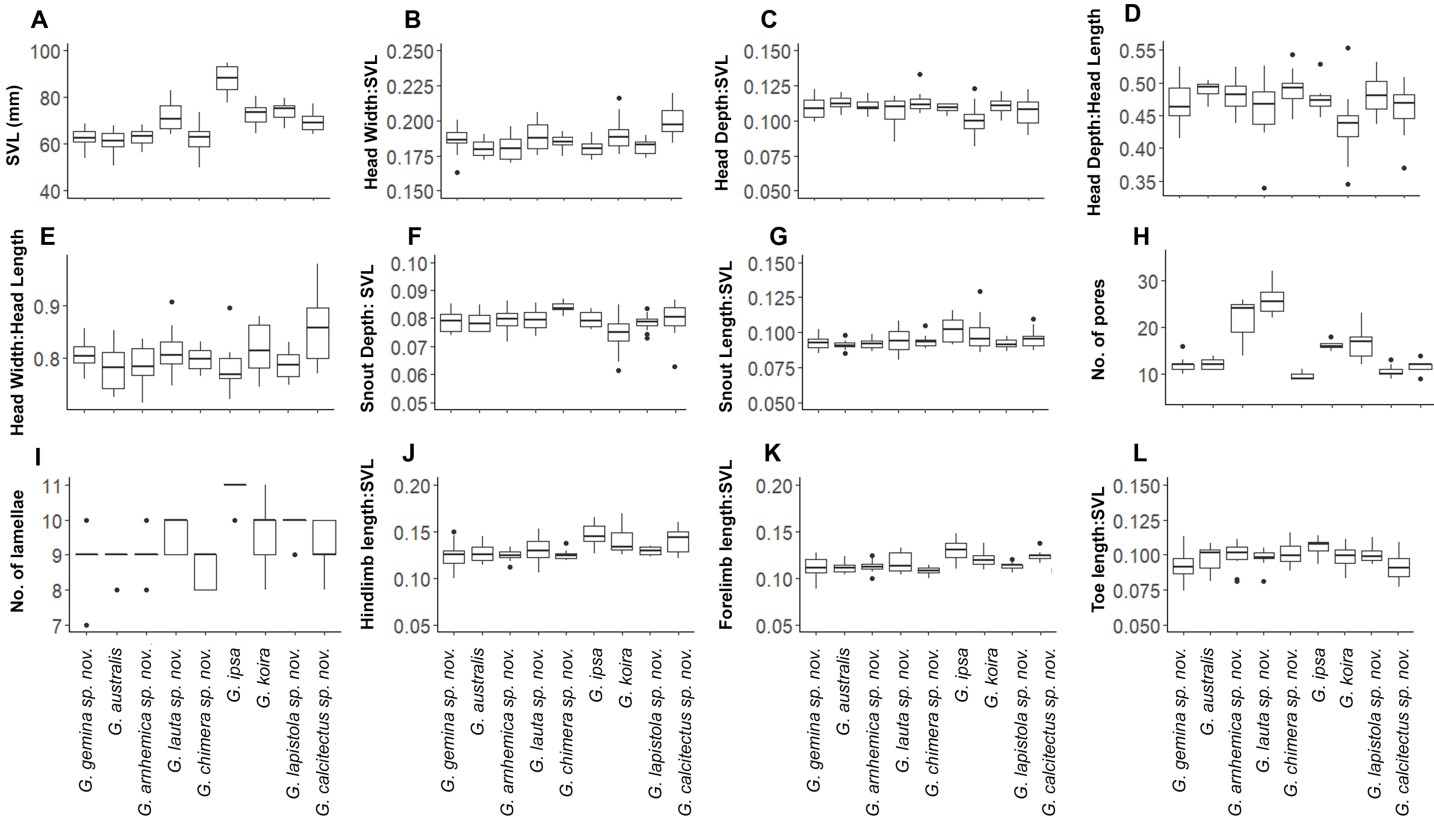

**Figure 5** Ranges for univariate morphological and meristic traits for species in the *Gehyra australis* and *G. koira* complexes. (A) Snout-vent Lenght (SVL); (B) Head Width/SVL; (C) Head Depth/SVL; (D) Head Depth/Head Length; (E) Head Width/Head Length; (F) Snout Depth/SVL; (G) Snout Length/SVL; (H) number of pores; (I) number of lamallae; (J) Hindlimb length/SVL; (K) Forelimb length; (L) Toe length/SVL.

(Fig. 8). Within the *G. koira* complex, *G. ipsa* and *G. koira* often had a bolder pattern on the head and dorsum including a moderately distinctive dark postorbital stripe, and light and dark dorsal banding or blotching (Fig. 9). *Gehyra calcitectus* sp. nov. was highly variable, but tended to have light ocelli (as opposed to bands), whereas *G. lapistola* sp. nov. tended to have little or no dorsal pattern (Fig 9; Table 2). *Gehyra chimera* sp. nov. tended have a light greyish dorsum with scattered darker brownish flecks and bands, very similar to most species in the *G. australis* complex (Fig. 10).

## Systematics of the *Gehyra australis* complex
### Summary assessment of species diversity and boundaries

The primary evidence for the presence of multiple species within *G. australis sensu lato* comes from the genetic analyses, with only subtle differences revealed in post hoc analyses of morphology. Moreover, there was little reason to suspect high species diversity in the *G. australis* complex based on observations of variation in morphology, pattern and behaviour by themselves, prior to the molecular genetic analyses (Oliver, Doughty and Moritz, 2007–2012, personal observations). The four candidate taxa within this complex were originally delineated using phylogenetic and coalescent-based analyses of an

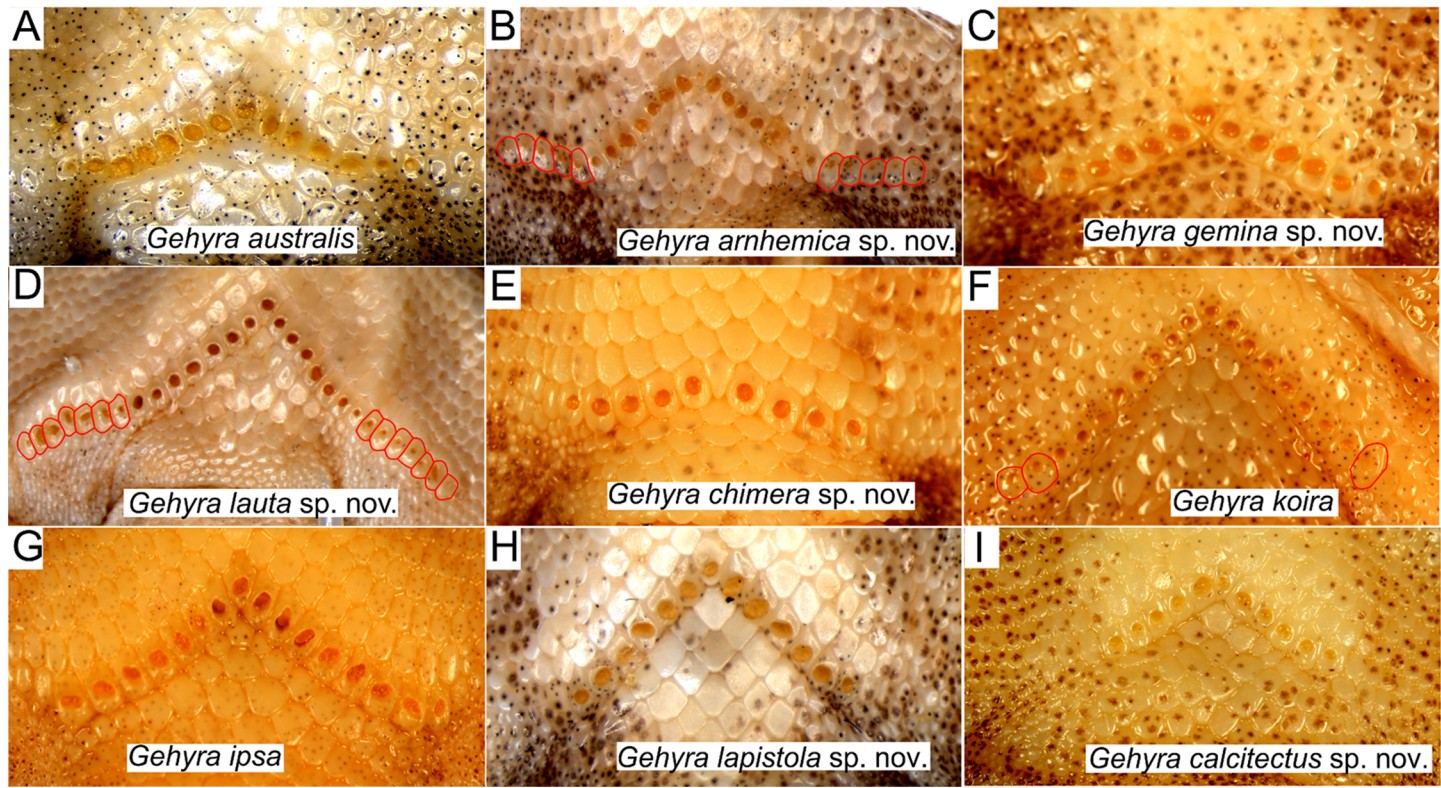

**Figure 6** Pre-cloacal pore arrangements in adult males of species in the *Gehyra australis* and *G. koira* complexes. (A) *G. australis* NTM R38170 (field # CCM7006); (B) *G. arnhemica* sp. nov. NTM R22626; (C) *G. gemina* sp. nov. WAM R172873; (D) *G. lauta* sp. nov. NTM R21311; (E) *G. chimera* sp. nov. WAM R177684; (F) *G. koira* WAM R164768; (G) *G. ipsa* WAM R101238; (H) *G. lapistola* sp. nov. NTM R37093; (I) *G. calcitectus* sp. nov. WAM R177691.         

extensive set of exonic sequences, albeit with small sample sizes per taxon. Having greatly expanded geographic sampling for nDNA SNPs to represent the full geographic ranges, including parapatric and overlap zones, we find strong genetic cohesion within lineages, clear genetic distinctiveness at geographic boundaries and, hence, strong evidence for lack of introgression between these taxa. There are slight (Fig. 4B; Table 2) but consistent morphological differentiation in some traits, especially pore number, and aspects of scalation and size. Strong concordance of mtDNA with nDNA means that the former can be used to genetically diagnose taxa where morphology is ambiguous (e.g. juveniles and females in many cases). One individual that may potentially be a hybrid was identified. This specimen was from the wall of a roadhouse outside the natural range of one putative parental (*G. australis*) and close to the limits if the distribution of the other (*G. gemina* sp. nov.) suggesting this is an anthropogenically-mediated, aberrant occurrence. On the basis of this evidence for genetic cohesion and evolutionary differentiation we recognise four evolutionarily distinct and cohesive lineages (species) within the *G. australis* complex.

## Nomenclatural history and application of names

The original description of *G. australis* was based on two specimens in the NHMUK (Gray, 1845). Subsequently *Cogger, Cameron & Cogger (1983)* designated NHMUK

**Table 2 Diagnostic morphological characters for species of the *Gehyra australis* complex and *G. koira* complex.**

| | Gehyra australis | Gehyra gemina sp. nov. | Gehyra arnhemica sp. nov. | Gehyra lauta sp. nov. | Gehyra chimera sp. nov. | Gehyra koira | Gehyra ipsa | Gehyra lapistola sp. nov. | Gehyra calcitectus sp. nov. |
|---|---|---|---|---|---|---|---|---|---|
| SVL | 61.5 | 62.9 | 62.9 | **72.2** | 62.5 | 72.5 | **87.9** | 74.2 | 69.4 |
| | (50.9–68.1) | (54.2–68.9) | (56.4–68.2) | **(64.4–83.1)** | (50.1–73.7) | (64.4–80.4) | **(77.6–94.9)** | (66.9–79.5) | (57.4–77.9) |
| Pre-cloacal pores | 13 | 12 | **24** | **26** | 10 | **17** | 16 | 10 | 12 |
| | (11–14) | (10–16) | **(21–26)** | **(22–32)** | (9–11) | **(12–23)** | (15–18) | (9–13) | (9–14) |
| HeadW/SVL | 0.180 | **0.188** | 0.180 | 0.183 | 0.185 | 0.190 | 0.183 | 0.181 | **0.199** |
| | (0.172–0.190) | **(0.163–0.204)** | (0.170–0.196) | (0.175–0.198) | (0.175–0.194) | (0.176–0.216) | (0.172–0.194) | (0.173–0.189) | **(0.183–0.219)** |
| HeadD/SVL | 0.113 | 0.109 | 0.110 | 0.111 | 0.113 | **0.100** | 0.109 | 0.111 | 0.107 |
| | (0.104–0.120) | (0.099–0.122) | (0.102–0.119) | (0.099–0.119) | (0.103–0.112) | **(0.081–0.123)** | (0.103–0.112) | (0.100–0.121) | (0.090–0.122) |
| Snoutd/SVL | 0.79 | 0.079 | 0.079 | 0.077 | **0.084** | 0.074 | 0.08 | 0.078 | 0.079 |
| | (0.074–0.085) | (0.074–0.085) | (0.072–0.086) | (0.074–0.083) | **(0.081–0.087)** | (0.061–0.085) | (0.076–0.084) | (0.073–0.084) | (0.063–0.085) |
| HindlimbL/ SVL | 0.128 | 0.125 | 0.125 | 0.120 | 0.129 | **0.141** | **0.148** | 0.130 | **0.142** |
| | (0.115–0.144) | (0.100–0.150) | (0.113–0.134) | (0.106–0.127) | (0.120–0.166) | **(0.125–0.169)** | **(0.127–0.165)** | (0.124–0.134) | **(0.123–0.162)** |
| Relative chinshield length | 0.63 (0.58–0.70) | 0.69 (0.56–0.92) | 0.67 (0.57–0.77) | **0.77** **(0.70–0.88)** | 0.75 (0.68–0.80) | 0.72 (0.64–0.79) | 0.78 (0.75–0.81) | 0.68 (0.62–0.74) | 0.66 (0.53–0.79) |
| Relative chinshield width | 0.78 (0.64–0.92) | 0.79 (0.61–0.84) | 0.79 (0.59–0.95) | 0.87 (0.73–1.01) | **0.85** **(0.61–0.95)** | 0.85 (0.69–1.08) | 0.79 (0.74–0.83) | 0.87 (0.77–0.96) | 0.77 (0.65–0.89) |
| Enlarged scale behind first chinshield pair | No | No | No | No | No | No | **Usually** | No | No |
| Pale ocelli on dorsum | No | No | No | No | No | No | No | No | **Yes** |
| Pale buff or brown transverse barring | No | No | No | No | Rarely | **Usually** | **Always** | No | No |
| Dark brown/ grey pattern on dorsum | Sometimes | Sometimes | Sometimes | **No** | **Often** | Rarely | Sometimes | **No** | Sometimes (eastern pops only) |
| Habitat | Trees, rocks buildings | Trees, buildings | Trees, rocks buildings | Trees | Trees, buildings | Rocks | Rocks | Rocks | Rocks (limestone) |

**Note:**
Key traits that help to distinguish species from other morphologically similar taxa are in bold.

xxii.55b (Port Essington, NT, Australia) as a lectotype and NHMUK xxii.551a (Swan River, WA, Australia) as a paralectotype (= *G. variegata* fide *Cogger, Cameron & Cogger (1983)*). Genotyped specimens from the type locality of *G. australis* (Port Essington, NT, Australia) are all positioned within the 'australis 2' clade, indicating 'australis 2' corresponds with true *G. australis* Gray, 1845. The lectotype of *G. australis* may also be further morphologically distinguished from *G. lauta* sp. nov. and *G. arnhemica* sp. nov. by its relatively shorter pore series (Fig. 10) (14 vs. 21–26, 22–32, respectively).

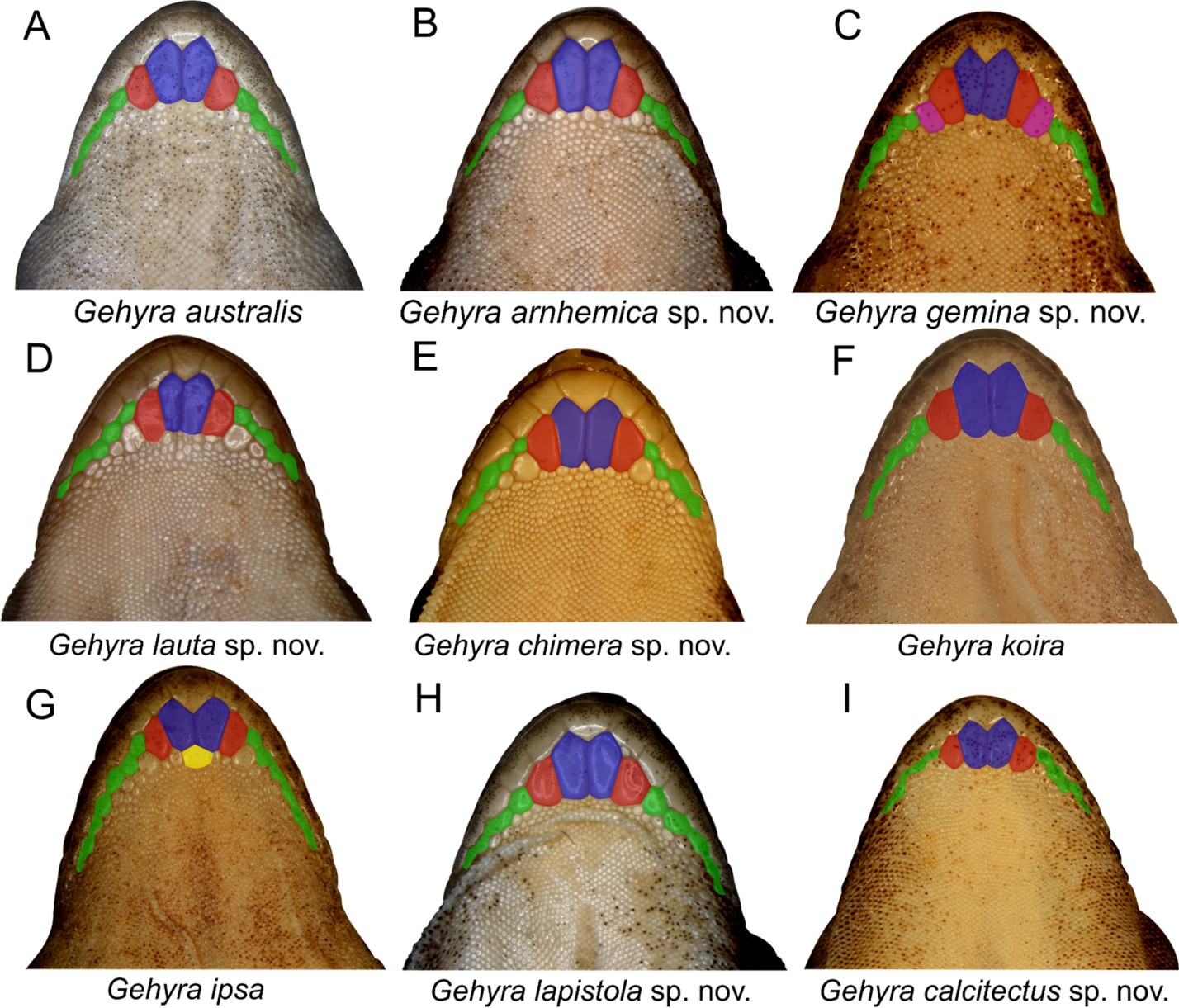

**Figure 7 Chin shield arrangements in for species in the *Gehyra australis* and *G. koira* complexes.** (A) *G. australis* NTM R38170 (field # CCM7006); (B) *G. arnhemica* sp. nov. NTM R22626; (C) *G. gemina* sp. nov. WAM R172873; (D) *G. lauta* sp. nov. NTM R21311; (E) *G. chimera* sp. nov. WAM R177684; (F) *G. koira* WAM R164768; (G) *G. ipsa* WAM R101238; (H) *G. lapistola* sp. nov. NTM R37093; (I) *G. calcitectus* sp. nov. WAM R177691.

We consider that there are no other names for the three additional clades in the *G. australis* complex identified from our analyses. *Phyria punctulata* Gray, 1842 has previously been treated as a synonym of *G. australis* and/or a *nomen oblitum*; however, *Ellis et al. (2018)* indicated the description of this form may not correspond with *G. australis* owing to an allusion to divided lamellae along with the meaning of the name which means 'small spots'. Instead, the description may be of a species within the *G. nana* group (i.e. mostly likely *G. nana* Storr or *G. paranana* Bourke, Doughty, Tedeschi, Oliver &

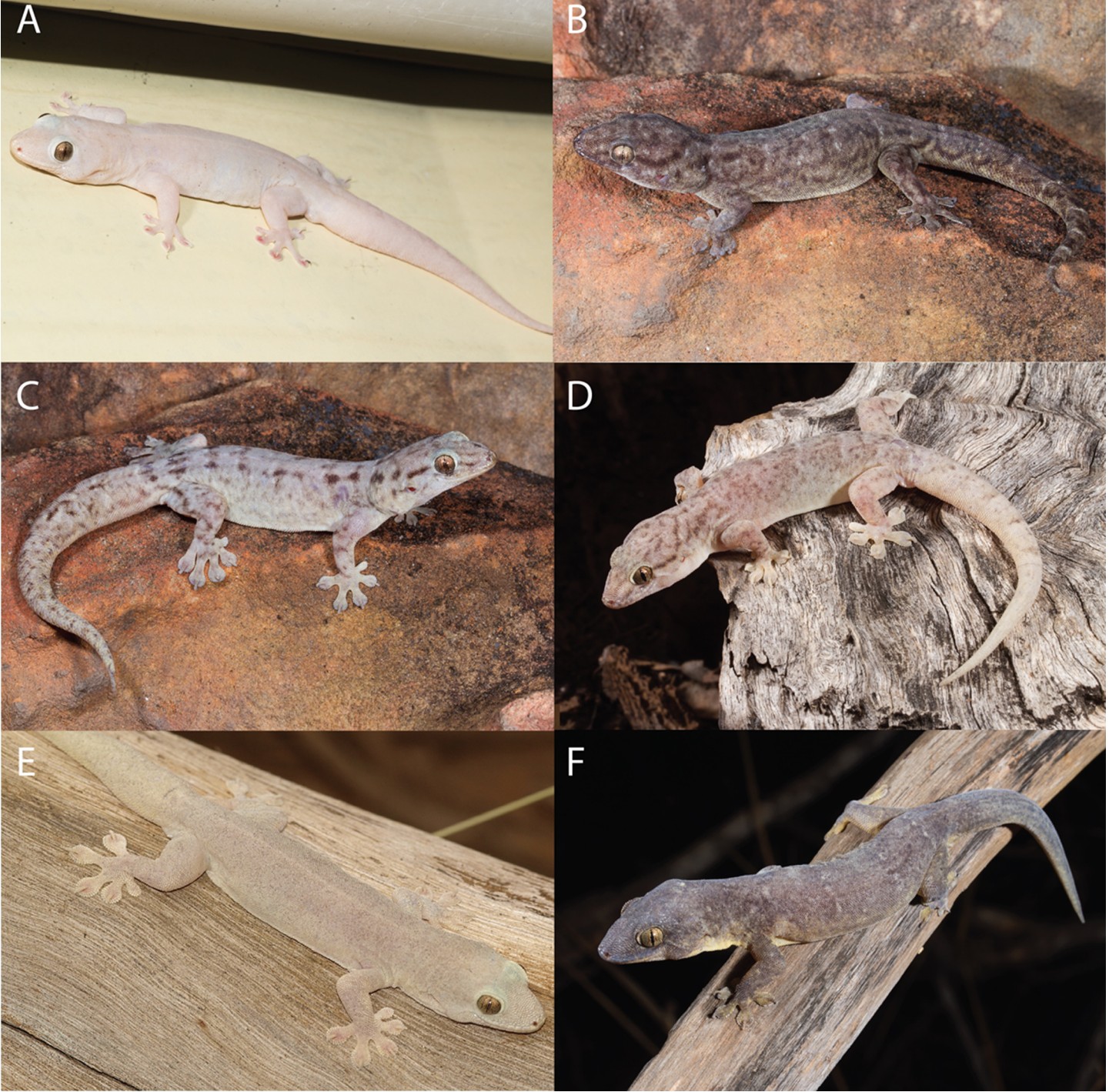

**Figure 8 Species in the *Gehyra australis* complex in life.** (A) *G. australis* Mary River Roadhouse, NT (Photo Credit: Tom Parkin); (B) *G. arnhemica* sp. nov. Wongalara Station, NT (Photo Credit: Stephen Zozaya); (C) *G. arnhemica* sp. nov. Wongalara Station, NT (Photo Credit: Stephen Zozaya); (D) *G. gemina* sp. nov. Halls Creek, WA (Photo Credit: Stephen Zozaya); (E) *G. lauta* sp. nov. Sybella Creek, QLD (QM J90707 holotype; Photo Credit: Mark Hutchinson); (F) *G. lauta* sp. nov. Calvert River, NT (Photo Credit: Stephen Zozaya).

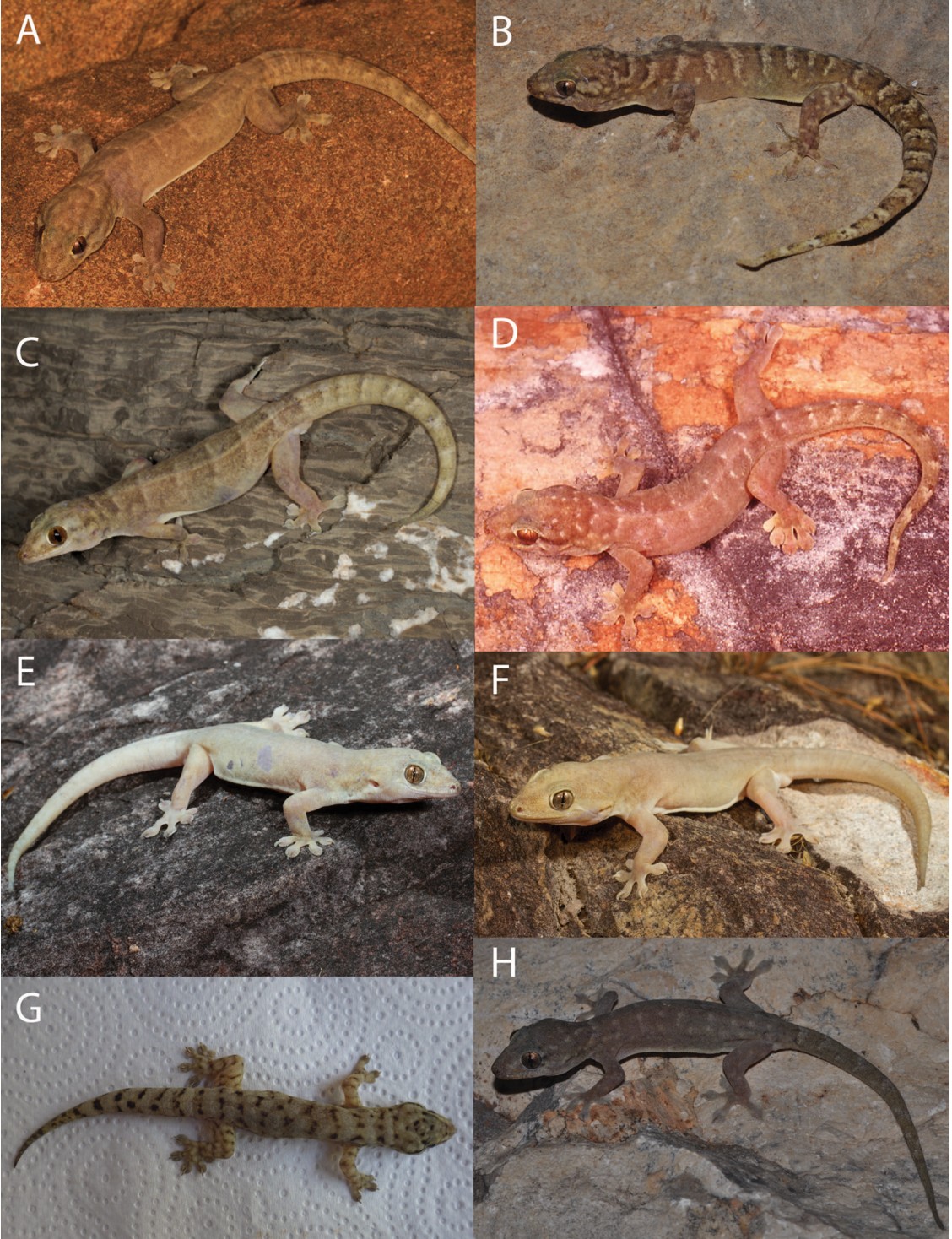

**Figure 9 Species in the *Gehyra koira* complex in life.** (A) *G. koira* Coucal Falls, WA (WAM R173932; Photo Credit: Ryan Ellis); (B) *G. koira* Gogo Station, WA (NMV D77028; Photo Credit: Paul Oliver); (C) *G. ipsa* Purnululu P, WA (Photo Credit: Ray Lloyd); (D) *G. ipsa* Purnululu NP, WA (Photo Credit: Brad Maryan); (E) *G. lapistola* sp. nov. Lost City, Litchfield NP, NT (Photo Credit: Chris Jolly); (F) *G. lapistola* sp. nov. Snake Creek Bunkers, Adelaide River Township, NT (Photo Credit: Chris Jolly); (G) *G. calcitetus* sp. nov. Argyle Station, WA (WAM R177704 paratype; Leonardo Tedeschi); (H) *G. calcitectus* sp. nov. Gogo Station, WA (WAM R177691 holotype; Photo Credit: Paul Oliver).

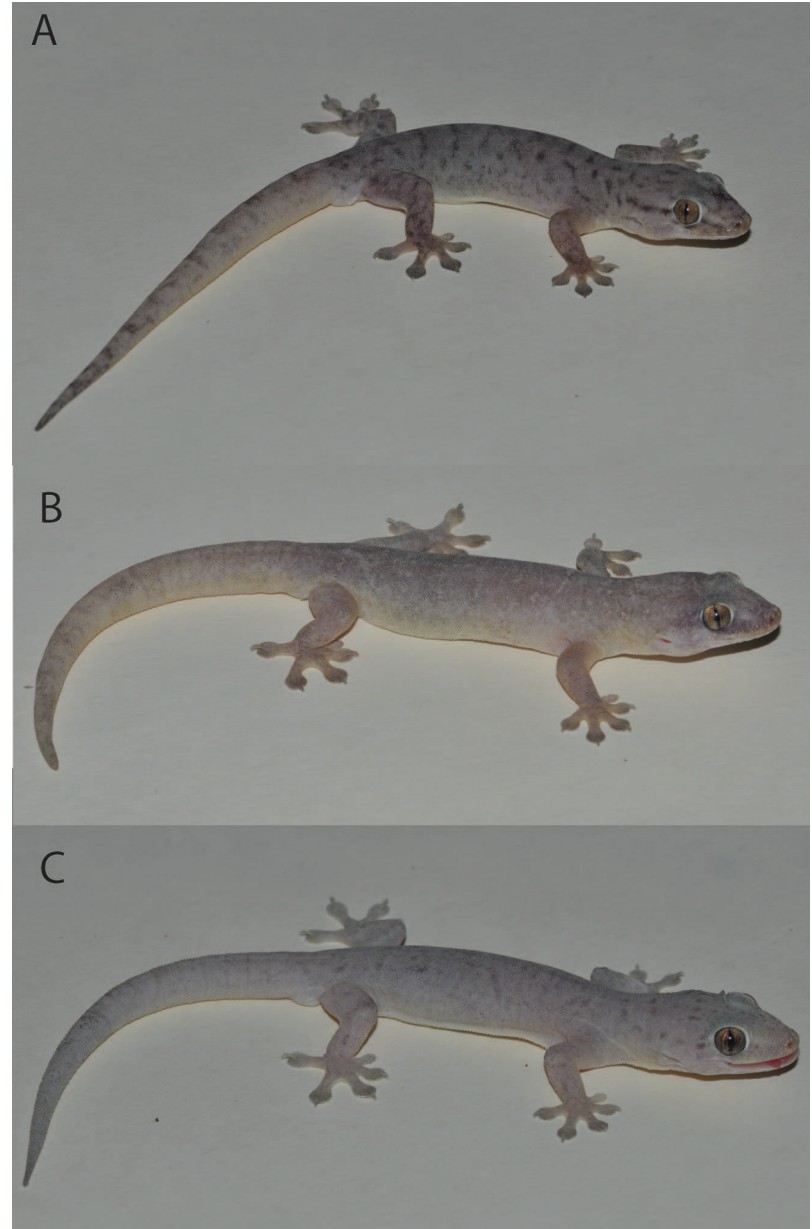

**Figure 10 *Gehyra chimera* sp. nov. and near sympatric *G. gemina* sp. nov. in life.** (A) *G. chimera* sp. nov. holotype (WAM R177687), Bell Creek Crossing, Gibb River Road, WA (WAM R177687), (B) *G. chimera* sp. nov. paratype (WAM R177686), same locality as holotype, (C) *G. gemina* sp. nov. (WAM R177732), Gibb River Road, Border of Wilinggin Conservation Park. (Photo Credit: Paul Oliver.)

Moritz) or a composite type series comprised of more than one *Gehyra* species. In the absence of a type specimen, it is not possible to confidently allocate *Phyria punctulata* to any known species with certainty, and the name is considered a *nomen oblitum* since it has not been in use since it was proposed in 1842 (*Ellis et al., 2018*). Accordingly, the name *Phyria punctulata* is not considered to be a name of relevance to any of the *G. australis* or *G. koira* complex taxa resulting from this study. *Gekko (Gehyra) grayi* has sometimes

been considered a synonym of *G. australis*; however, *Cogger, Cameron & Cogger (1983)* placed this name in the synonymy of eastern populations of *G. variegata* (i.e. now *G. versicolor Hutchinson et al., 2014*), and it is not considered relevant to any lineages in the *G. australis* group. Following *Kaiser et al. (2013)* and an official statement from the *Australian Society of Herpetologists (2016*: accessed 28 April 2019*)* we do not consider nomenclatural acts pertaining to the Australasian herpetofauna that have appeared outside the peer-reviewed literature.

### *Gehyra australis* complex diagnosis and description

A group of medium to moderately-large sized *Gehyra* (max SVL 68.1–83.1 mm), torso slightly dorsoventrally compressed with fine homogenous rounded scales on dorsum and flattened scales on ventrum, snout moderately long with rounded tip and covered with enlarged rounded scales, eyes large and protruding, ear opening small, rostral wide and in contact with nostrils, large internasals bordering nares separated by 0–3 smaller internasals of greatly varying size, nostril in contact with rostral scale, limbs short with claws on digits II–V, claws protruding from dorsal surface of expanded toe pad, hindlimbs without a posterior skinfold, subdigital lamellae under fourth toe undivided or at most with shallow indistinct groove, tail cylindrical tapering to a fine point, base colouration of dorsum in life usually grey and either plain without pattern, or with fine vermiculations and scattered spots, in males 9–32 pre-cloacal pores in shallow chevron, and in females two eggs per clutch.

### *Gehyra australis* Gray, 1845
Western Top End Gehyra
aus2 of *Noble et al. (2018)*
aus2 of *Oliver et al. (2019)*
Figs. 6–8A, 11 and 12

*Lectotype.* NHMUK xxii.55b, adult male, from Port Essington, NT *fide* Gray (1845) (−11.36°S, 132.15 (approximate co-ordinates inferred from Google Earth)) (Fig. 11).

*Paralectotype.* NHMUK xxii.55a, Swan River, WA (= *Gehyra variegata fide* Cogger, Cameron & Cogger, 1983).

*Referred material.* See Tables S1 and S2.

*Diagnosis.* A large *Gehyra* species (up to 68.1 mm SVL), differing from all other *Gehyra* species outside of the *G. australis* complex as per the diagnosis above. Differs from other members of the *G. australis* complex in the combination of: moderate size within complex (adult SVL up to 68.1 mm, mean 62.7 mm); pre-cloacal pores in males not numerous (11–14), mostly equal in size, not extending onto limbs and not distinctly tapering in size distally; suture between first and second chin shields usually straight; second chin shields approximately two thirds length of first chin shields (mean ratio 0.63, range 0.58–0.70); and base colouration of adults smoky grey to brown, often with faint but extensive pattern of indistinct darker grey or brown vermiculations across the head,

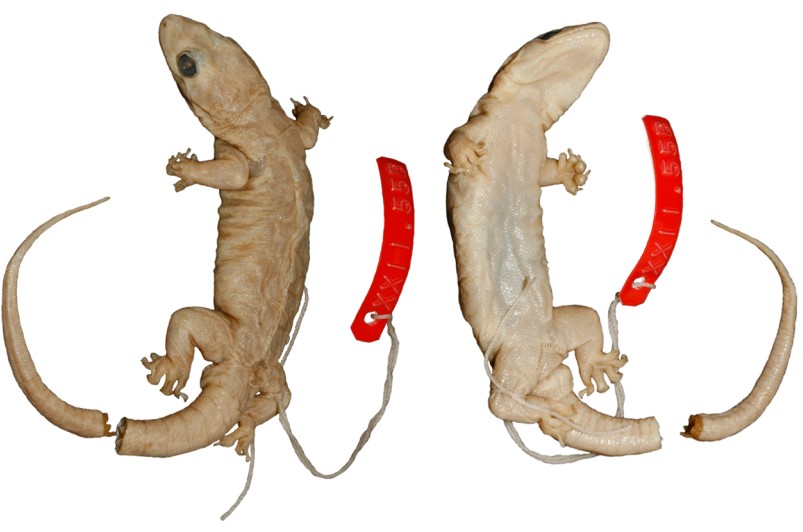

**Figure 11 Lectotype specimen of *Gehyra australis* (NHMUK xxii.55b).**

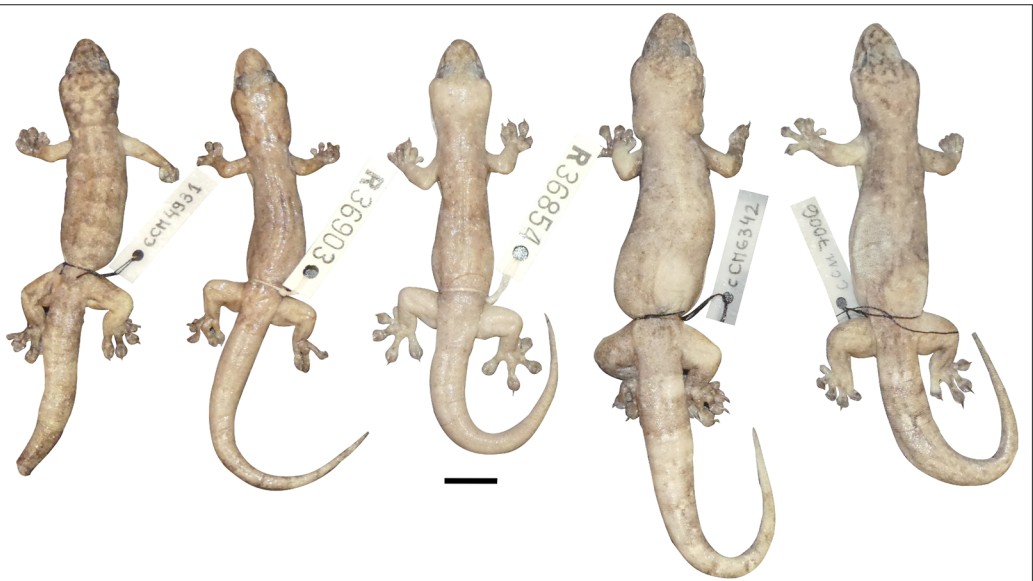

**Figure 12 Variation amongst preserved *Gehyra australis*.** From left to right, NTM R38168 (field # CCM4981), NTM R36903, NTM R36854, NTM R38197 (field # CCM6342), NTM R38170 (field # CCM7006). Scale bar = 10 mm.

body and tail. Further diagnosed from other species within the *G. australis* complex genetically by three unique amino acids in the *ND2* locus (Table 1).

*Gehyra australis* may occur in close geographic proximity to two other members of the *G. australis* group; *G. lapistola* sp. nov. and *G. pamela*. *G. australis* differs from *G. lapistola* sp. nov. by its smaller size (mean and maximum adult SVL, respectively: 62.7 mm and 68.1 mm vs. 74.2 mm and 79.5 mm) and also tending to have more extensive dorsal patterning of vermiculations and flecking (vs. plain grey or brown with

no or very little pattern); and from *G. pamela* in lacking prominent pale spots and ocelli (vs. present), having a rounded snout tip in dorsal aspect (vs. squarish), and in having smaller chin shields (extending to approximately level with posterior edge of second infralabial vs. approximately level with posterior edge of third infralabial) (see *King (1982)* for images), and in having a lower number of pre-cloacal pores in adult males (11–14 vs. 18–24).

Within the *G. australis* complex, *G. australis* occurs in contact or in potential sympatry with *G arnhemica* sp. nov. and *G. gemina* sp. nov. *G. australis* differs from *G. arnhemica* sp. nov. in having a lower number of pre-cloacal pores in males (10–14 vs. 21–26), and also in tending to have less distinct and extensive dorsal patterning in life (faint barring vs a clear network of vermiculations). *G. australis* differs from *G. gemina* sp. nov. in having posterior edge of first infralabial generally ∼50% or greater the length of second supralabial (vs. ∼60% or less) and outer edge of first pair of chin shields in contact with second pair usually strait (vs. usually convex). It differs from *G. lauta* sp. nov. in its smaller size (mean and maximum adult SVL, respectively: 62.7 mm and 68.1 mm vs. 71.4 mm and 83.1 mm), second chin shields usually less than three-quarters length of first chin shields (mean and range ratios 0.63 (0.58–0.70) vs. 0.77 (0.70–0.88)), and fewer pre-cloacal pores (10–14 vs. 22–32) generally not extending onto limbs.

Based on the morphological characters we have examined, *G. australis* is most similar morphologically to *G. arnhemica* sp. nov. (particularly weakly patterned females), *G. gemina* sp. nov. (both sexes) and *G. chimera* sp. nov. (both sexes) of the *G. koira* complex. The relatively disjunct distributions (particularly *G. chimera* sp. nov.) permit identification in most cases when accurate locality data is available. Along the southern edge and central portions of the Top End region where some of these species may occur in sympatry, genetic data may be required to confidently identify specimens to species (see Table 1 for diagnostic amino acids).

*Description.* As for *G. australis* complex description treated herein above, with the differences and variation outlined in the diagnoses above and Table 2.

*Colour and pattern.* In life, colouration smoky grey to brown, often with scattered indistinct darker brown or darker grey flecks, spots, vermiculations or transverse bars extending across the dorsal and lateral surfaces of head, body, limbs and tail. Specimens found under lights and/or on light substrates (such as the wall of houses) tend be pink and to show very faint or no patterning (Fig. 8A), but often become darker when placed in a darker environment (P.M. Oliver, 2013, personal observations). In preservative, base colouration light to medium grey, sometimes with a brownish tinge, pattern generally absent or minimal and always indistinct, at most consisting of weak medium grey transverse bands, but more usually of very indistinct grey mottling and/or very tiny light ocelli.

*Distribution, habitat and ecology.* Known only from the NT, concentrated in the western portion of the Top End region, extending from Port Essington west through Darwin and surrounds, then along the western seaboard, as far south as Gregory NP, and east to at

Eva Valley on the southern edge of the Arnhem Plateau (Fig. 2). There are very few records of *G. australis* (s.l.) from the Arnhem Plateau itself and no specimens from there have tissues for genotyping.

Gehyra australis occurs on trees in both rocky areas and on open plains and woodlands. On trees it is most commonly observed on the lower trunks or large branches between 0 m and 4 m from the ground. It occurs rarely on rocky microhabitats, and then in areas where *G. lapistola* sp. nov. or *G. pamela* are absent, such as sandstone country around Hayes Creek and on both limestones and sandstones around Katherine (P.M. Oliver, 2013, personal observations). It has also been observed on human structures such as buildings (under lights) and concrete bridges and causeways. Anecdotal reports suggest that it may have been displaced from some anthropogenic habitats by *Hemidactylus frenatus* Dumeril & Bibron (*Greer, 1989*).

### *Gehyra arnhemica* sp. nov.

urn:lsid:zoobank.org:act:33DB9200-D783-4738-9B5A-50A6AF4155C1
East Arnhem Land Gehyra
aus3 of *Noble et al. (2018)*
aus3 of *Oliver et al. (2019)*
Figs. 6–8B, 8C and 13

*Holotype*. NTM R22626, adult male, collected from Long Billabong, Savannah Way, 2 km N Cox River, Roper Gulf, NT (−15.3067°S, 135.3408°E), collected by P. Horner, 19 May 1996.

*Paratypes* (*N* = 12). *Northern Territory*: NTM R38171 (field # CCM2271), NTM R38172 (field # CCM2272), Lake Katherine, NT (−14.3079, 135.0610°E); NTM R38173 (field # CCM2408), Ngukurr, NT (−14.6550°S, 134.7812°E); NTM R38174 (field # CCM2573), NTM R38175 (field # CCM2575), NTM R38176 (field # CCM2576), Cox River area, Limmen NP, NT (−15.3167°S, 135.3409°E); NTM R38178 (field # CCM6511), NTM R38179 (field # CCM6516), Emu Springs, NT (−13.1562°S, 134.8506°E); NTM R38180 (field # CCM6554), NTM R38181 (field # CCM6555), NTM R38182 (field # CCM6595), 3.0 km SW Gikal, NT (−12.0813°S, 136.2942°E); NTM R38183 (field # CCM6647), 1.5 km NE Gikal, NT (−12.0522°S, 136.3210°E).

*Referred material*. See Tables S1 and S2.

*Diagnosis*. A large *Gehyra* species (up to 68.2 mm SVL) differing from all other *Gehyra* species outside of the *G. australis* complex as per the complex diagnosis above. Differs from other members of the *G. australis* complex in the combination of: moderate size within complex (adult SVL up to 68.2 mm, mean 63.0 mm); pre-cloacal pores in males numerous (21–26), often extending onto limbs and reduced in size distally; suture between first and second chin shields usually straight; second chin shields approximately two thirds length of first chin shields (mean ratio 0.67, range 0.57–0.77); and base colouration of adults smoky grey to brown, often with distinct and extensive pattern of darker-brown vermiculations, stripes, scalloping and/or blotches across the head, body and tail.

 

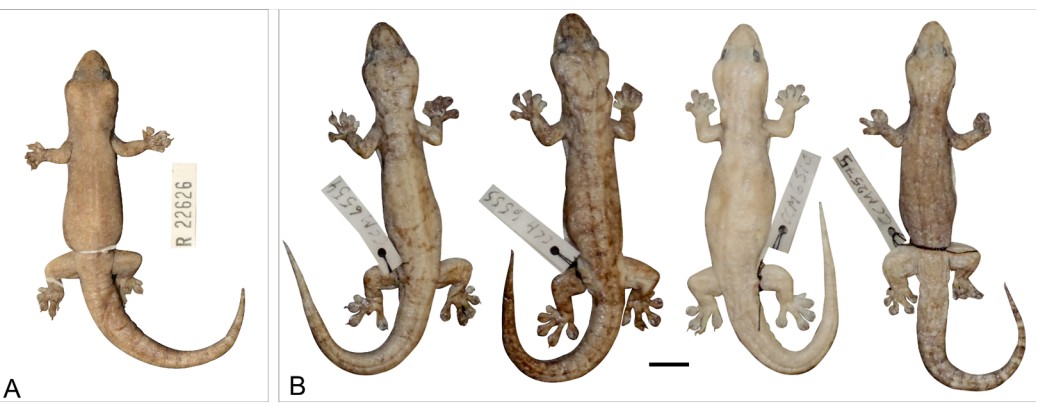

**Figure 13 Variation amongst preserved *Gehyra arnhemica* sp. nov. specimens.** From left to right, (A) holotype (NTM R22626) and (B) four paratypes (NTM R38180, NTM R38181, NTM R38179, NTM R38175). Scale bar = 10 mm.

Further diagnosed from other species within the *G. australis* complex genetically by three unique amino acids in the *ND2* locus (Table 1). Some specimens, especially juveniles and females, may only be diagnosable from other members of the *G. australis* complex and *G. chimera* sp. nov. on the basis of locality and/or genetic data.

*Gehyra arnhemica* sp. nov. overlaps or contacts the distribution of *G. australis* and *G. gemina* sp. nov. and is very similar morphologically to both species; however, males of *G. arnhemica* sp. nov. differ from both in having a higher number of pre-cloacal pores (21–26 vs. 11–14 in *G. australis* and 10–16 in *G. gemina* sp. nov.). It further differs from *G. gemina* sp. nov. in outer edge of first pair of chin shields in contact with second pair usually strait (vs. usually convex). *Gehyra arnhemica* sp. nov. further tends to differ from *G. australis* in having a more extensive and bolder pattern of dark brown blotches, lines and/or scalloping on the head, torso and tail. From *G. lauta* sp. nov., it differs in its smaller size (mean and maximum adult SVL, respectively: 68.2 mm and 63.0 mm vs. 71.4 mm and 83.1 mm).

*Gehyra arnhemica* sp. nov. also overlap or abuts with the distribution of the rock-dwelling taxa *G. borroloola* and *G. pamela*; however, it can be readily differentiated from both by the absence of prominent pale spots and ocelli (vs. present), and by having smaller chin shields (extending to approximately level with posterior edge of second infralabial vs. approximately level with posterior edge of third infralabial) (see *King, 1982*).

*Description*. As for *G. australis* complex description above, with the differences and variation outlined in the diagnoses above and in Table 2.

*Colour and pattern*. Recently deceased (Gikal, North-east Arnhem Land, NT, Australia) and live specimens (Wongalara Station) have pale brown-grey dorsal surface with unaligned dark brown streaks, paired blotches or extensive vermiculations, sometimes bordered by paler margins across the dorsum (Figs. 8B and 8C). Head and limbs likewise greyish, again with some to extensive dark brown spots and/or striping, including a moderately prominent and clearly defined postorbital stripe. Head also with occasional

pale spots. Original tails with some to extensive dark brown patterning. In preservative, dorsal colouration light to medium grey, often with transverse darker grey bands or series of blotches along the dorsum and tail, grey bands sometimes with faint off-white margins, especially on tail, further dark grey flecks, splotches of maculations present across the dorsal surfaces, some specimens also show very indistinct off-white dorsal maculations. Ventral colouration off-white with or without pale grey maculations, especially around the lateral extremities and on the throat.

*Summary description of holotype* (*NTM R22626*). All measurement in mm: SVL 63.2; TrunkL 23.3; TrunkW 10.7; ForelimbL 6.9; HindlimbL 7.9; HeadL 14.2; HeadD 7.2; HeadW 11.5; SnoutL 5.8; SnoutD 5.2; ToeL 6.4. Rostral flat and rectangular, with slightly rounded dorsal corners and medial crease on dorsal half. Nostrils separated by two large internasals; supralabials to mid-point of eye 8 on both sides. Infralabials 9 on both sides; mental scale triangular, chin shields elongate, rounded and in two pairs; parainfralabials rounded, enlarged and varying from round to oblong. Limbs relatively short, with nine divided subapical subdigital lamellae undivided under fourth right toe. Tail original. Pre-cloacal pores 25, arranged in a broad chevron formation.

*Distribution, habitat and ecology*. Distribution centred on the eastern portion of the Top End region of the NT, including offshore islands to the north (Inglis Island) and east (Groote Eylandt), south as far as Limmen NP, and west to the upper Jalboi River (Fig. 2). There is also an isolated record from Gudjekbin in north central Arnhem Land, suggesting that this species occurs more widely in this relatively poorly sampled region.

Occurs on both arboreal and rocky microhabitats. At Wongalara station in southern Arnhem Land they have been recorded from smooth-barked Eucalypt trees (*Corymbia polycarpa*) in seasonally inundated woodlands, and were also not found on *Melaleuca* sp. trees in the same habitat (J. Smith, 2013, personal communication). At Emu creek outstation (central Arnhem Land) they were on smooth-bark Eucalypt trees and human structures, while in north-east Arnhem Land (e.g. around Mata Mata) they were abundant on large rock boulders and none were seen on trees, although no smooth barked trees were observed at these sites (C. Moritz, 2016, personal observations).

*Etymology*. The species epithet refers to the Arnhem Land region of the north-east Top End of the NT, in reference to the species occurrence and apparent endemism to the region.

### *Gehyra gemina* sp. nov.
urn:lsid:zoobank.org:act:BC6CA755-1F3D-4E03-9A09-8E2F1174CC09
Plain Tree Gehyra
aus1 of *Noble et al. (2018)*
aus1 of *Oliver et al. (2019)*
Figs. 6–8D, 10C and 14

*Holotype*. WAM R179349 (field # CCM3042), adult male, collected from Saw Tooth Gorge, WA (−18.4252°S, 127.8197°E), collected by A.C. Alfonso Silva, M. Pepper and S. Potter, 11 July 2014.

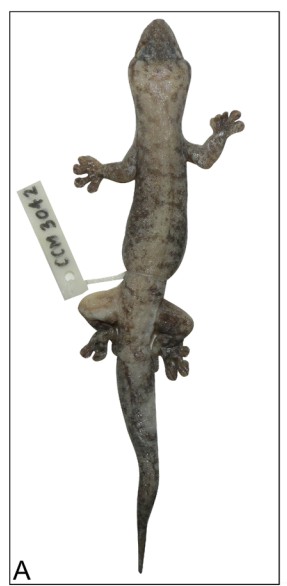
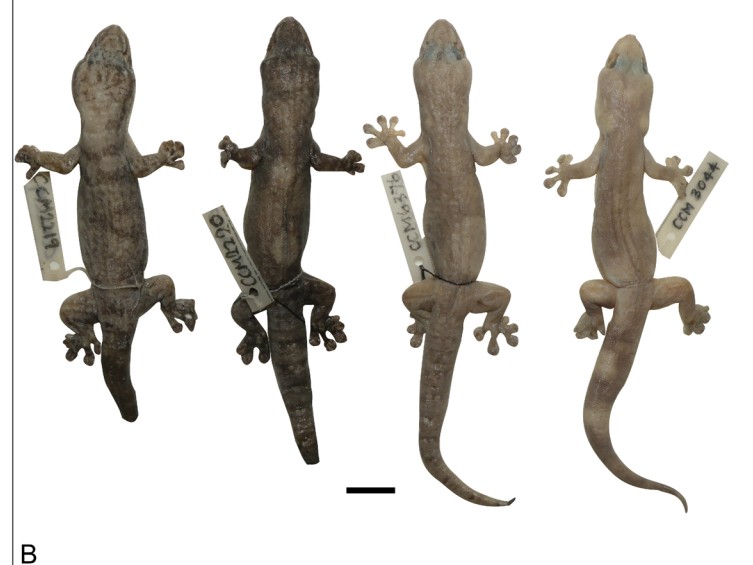

**Figure 14 Variation amongst preserved type series of *Gehyra gemina* sp. nov.** (A) From left to right, (A) holotype (WAM R179349) and (B) four paratypes (NTM R38158, NTM R38159, NTM R38164, NTM R38163). Scale bar = 10 mm.

*Paratypes (N = 20). Western Australia:* WAM R177723 (field # CCM3221), Muluk rest area, 39 km SSW Warmun, WA (−17.3389°S, 128.0525°E); WAM R177725 (field # CCM3224), WAM R177726 (field # CCM3225), Purnululu NP access road, WA (−17.4203°S, 128.0922°E); WAM R177728 (field # CCM3298), Gogo Station, WA (−18.4731°S, 125.8171); WAM R177732 (Field # CCM3374), Gibb River Road, Wilinggin Conservation Park, WA (−17.15351°S, 125.42159°E); WAM R177734 (field # CCM7280), WAM R17735 (field # CCM7281), Lissadell Station, WA (−16.6635°S, 128.5265°E); WAM R177737 (field # CCM7303), Mt. Nyulasy, WA (−16.7655°S, 128.2730°E). *Northern Territory:* NTM R38152 (field # CCM0367), Calvert Hills Station, NT (−17.1979°S, 137.4343°E); NTM R38153 (field # CCM0404), McArthur River Station, NT (−16.6383°S, 135.8462°E); NTM R38154 (field # CCM0406), McArthur River Station, NT (−16.6505°S, 135.8493°E); NTM R38155 (field # CCM0554), Gregory NP, NT (−15.6109°S, 131.1160°E); NTM R38157 (field # CCM2218), NTM R38158 (field # CCM2219), NTM R38159 (field # CCM2220), Munbililla (Tomato Island) Limmen NP, NT (−14.7794°S, 134.6667°E); NTM R38160 (field # CCM2479), Butterfly Springs, Limmen NP, NT (−15.62723°S, 135.4623°E); NTM R38161 (field # CCM2612), Southern Lost City, Limmen NP, NT (−15.6272°S; 135.4623°E); NTM R38162 (field # CCM2618), Fossil Fern, Lorella Springs Station, NT (−15.6795°S, 135.6247°E); NTM R38163 (field # CCM3044), Limbunya Station, NT (−17.5189°S, 129.8759°E); NTM R38164 (field # CCM6376), Mallapunya Station, NT (−16.9732°S, 135.8030°E).

*Referred material.* See Tables S1 and S2.

*Diagnosis.* A large *Gehyra* species (up to 68.9 mm SVL), differing from all other *Gehyra* species outside of the *G. australis* complex as per the diagnosis above. Differs from

other members of the *G. australis* complex in the combination of: moderate size within complex (adult SVL up to 68.9 mm, mean 62.9 mm); pre-cloacal pores in males not numerous (10–16), generally equal in size, not extending onto limbs and not distinctly tapering in size distally from largest median pore, distal most pores ~equal to no smaller than one-half the size of median pore; outer edge of first chin shield concave where it contacts the second chin shield, rarely straight; second chin shields approximately two-thirds length of first chin shields (mean ratio 0.64, range 0.56–0.75); and base colouration of adults smoky grey to brown, often with faint but extensive pattern of indistinct darker grey or brown vermiculations across the head, body and tail. If transverse lines present on dorsum, lines are irregular and without strait edges.

Further diagnosed from other species within the *G. australis* complex genetically by two unique amino acids in the *ND2* locus (Table 1).

The geographic range of *G. gemina* sp. nov. overlaps or contacts the distribution of all three other species in the *G. australis* complex; it differs from *G. lauta* sp. nov. in its smaller size (mean and maximum adult SVL respectively: 62.1 mm and 68.9 mm vs. 71.4 mm and 83.1 mm), second chin shields usually less than three-quarters length of first chin shields (mean and range ratios 0.69 (0.56–0.75) vs. 0.77 (0.70–0.88)), and fewer pre-cloacal pores (10–16 vs. 22–32), not extending onto limbs and of relatively similar size; from *G. arnhemica* sp. nov. by fewer pre-cloacal pores in males (10–16 vs. 21–26) not extending onto limbs and outer edge of first pair of chin shields in contact with second pair usually convex, rarely strait (vs. usually strait); and from *G. australis* by posterior edge of first infralabial generally ~60% or less the length of second supralabial (vs. generally 50% or greater) and outer edge of first pair of chin shields in contact with second pair usually convex, rarely strait (vs. usually strait).

*Gehyra gemina* sp. nov. is morphologically most similar to *G. arnhemica* sp. nov. (particularly juveniles and females), *G. australis* (both sexes) and *G. chimera* sp. nov. (both sexes) of the *G. koira* complex, and genetic data may be necessary to identify individuals with certainty, particularly in areas of sympatry or parapatry.

*Description.* As for *G. australis* complex description above, with the differences and variation outlined in the diagnoses above and in Table 2.

*Colour and pattern.* Specimens photographed in life from the Kimberly (King Leopold Ranges, Gogo Station, Halls Creek, WA, Australia) show a pale off-white to grey dorsal surface, with variable amounts of darker brown or grey streaks, flecks or vermiculations, and occasionally very indistinct whitish dorsal bands (Fig. 8D). When transverse bands present, lines are irregular and without strait edges. Head and limbs likewise greyish, again with some to extensive brown or darker grey spots and or striping, including a moderately to weakly defined postorbital stripe. Original tails also with some to extensive darker brown patterning. In preservative, dorsal background colouration varies from light to dark grey, sometimes plain and unpatterned, but sometimes with dark grey vermiculations and blotching on the head and/or across the dorsum and tail. Ventral surfaces of body pale cream with no extensive grey maculations. Subcaudal surfaces usually pale cream.

*Summary description of holotype* (*WAM R179349*). All measurements in mm: SVL 62.1; TrunkL 27.8; TrunkW 10.5; ForelimbL 7.7; HindlimbL 8.3; HeadL 14.2; HeadD 6.4; HeadW 11.5; SnoutL 6.1; SnoutD 4.7; ToeL 5.1.

Rostral broadly rectangular, with flat dorsal edge and rounded corners; medial crease on upper half; internasals 4; supralabials (to mid-point of eye) 8; infralabials 8 (left) 9 (right); mental scale triangular, chin shields elongate, rounded and in two pairs. Body slightly compressed. Limbs relatively short, subdigital lamellae undivided, 10 on fourth right toe. Tail regrown. Pre-cloacal pores 10, ~equal in size, arranged in a broad chevron formation, not extending onto limbs.

*Distribution, habitat and ecology.* Occurs widely through the northern deserts of WA and the NT. Occurs from Broome in the west, east throughout the Kimberley region of WA, north to Victoria River and the southern edge of the Arnhemland region, and throughout the Gulf Country in the NT (Fig. 2). The apparent hiatus of *G. gemina* sp. nov. in the central portion of the northern desert region may be artefact of poor sampling across the region; however, a shallow genetic disjunction across this relatively arid and sparsely vegetated area (e.g. the two clusters in the PCoA; Fig. 3A) indicates that the absence of records may represent true disjunction between populations.

Generally found on trunks or large branches of trees with few, if any, records from rocky microhabitats. Also found on buildings. Some isolated populations, usually as commensals may be introduced especially to the southern extremities of the range (see results).

*Etymology.* The species epithet is from the Latin word *gemina* (twin, same), in reference to the species' morphological similarities shared with other members of the *G. australis* complex, *G. australis* in particular.

*Remarks.* The wide distribution of this form across the southern portions of the AMT broadly overlaps with the northern deserts region (sensu *Cracraft, 1991*; but see *González-Orozco et al., 2014*). This distribution also mirrors that of several other species and lineages that have likewise only been identified recently, supporting the hypothesis that the southern AMT has a distinct associated endemic biota (*Smith et al., 2011*; *Catullo et al., 2014*; *Laver et al., 2017*).

### *Gehyra lauta* sp. nov.
urn:lsid:zoobank.org:act:02B9D2DC-9499-41E3-BD1E-D0F0C8880F83
Gulf Tree Gehyra
aus4 of *Noble et al. (2018)*
aus4 of *Oliver et al. (2019)*
Figs. 6–8E, 8F and 15

*Holotype.* QM J90707, adult male, collected from Sybella Creek, Dajarra Road, 17.2 km S Mount Isa, Qld (−20.876°S, 139.458°E), collected by M.N. Hutchinson, P.M. Oliver, M.A. Cowan and D.L. Rabosky, 20 April 2010.

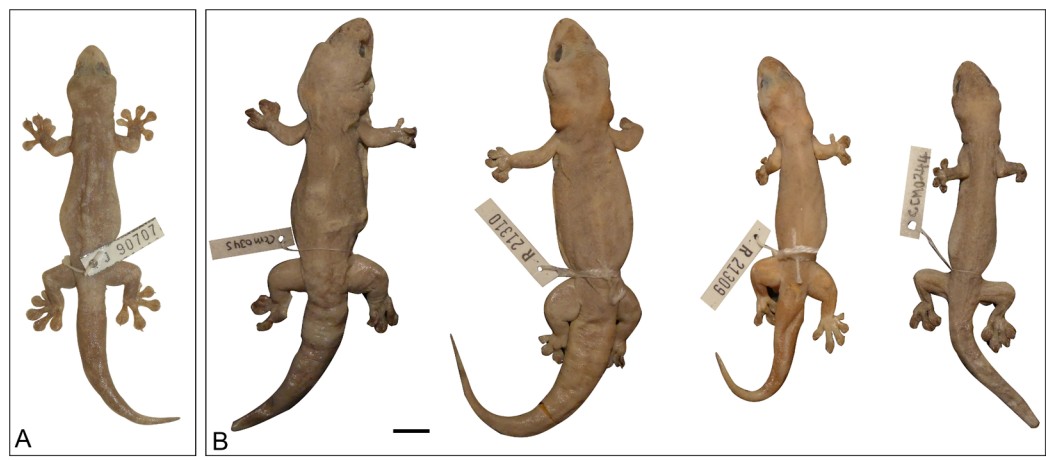

**Figure 15 Variation amongst preserved type series of *Gehyra lauta* sp. nov.** From left to right, (A) holotype (QM J90707) and (B) four paratypes (QM J96624, NTM R21310, NTM R21309, QM J96621). Scale bar = 10 mm.

*Paratypes (N = 14). Northern Territory*: NTM R38184 (field # CCM0438), MacArthur Station, NT (−16.7101°S, 136.2042°E). *Queensland*: QM J96621 (field # CCM0244), Gunpowder Ridge, Qld (−19.7503°S, 139.3808°E); QM J96622 (field # CCM0247), West Leichardt Station, Qld (−20.6027°S, 139.6839°E); QM J96623 (field # CCM0322), Bowthorn Station tip, Qld (−18.0938, 138.3017°E); QM J96624 (field # CCM0345), Kingfisher Station Qld (−17.8854°S, 138.2810°E); NTM R21309–12,Musselbrook Reservoir, Qld (−18.592°S, 138.124°E); QM J47904 Elizabeth Gorge, Bowthorn Station, Qld (18.216°S, 138.333°E); QM J75291 Lawn Hill NP, Qld (−18.692°S, 138.491°E); QM J78786, 10 km WNW Kajabbi, Qld (−19.776°S, 139.896°E); QM J88487, Mount Isa area, Qld (−20.514°S, 139.461°E); QM J90711, Sybella Creek, Dajarra Road, 17.2 km S Mount Isa, Qld (−20.876°S, 139.458°E).

*Referred material.* See Tables S1 and S2.

*Diagnosis.* A large *Gehyra* species (up to 83.1 mm SVL), differing from all other *Gehyra* species outside of the *G. australis* complex as per the diagnosis above. Differs from other members of the *G. australis* complex in the combination of: large size within complex (adult SVL up to 83.1 mm, mean 72.2 mm); pre-cloacal pores in males numerous (22–32), often extending onto limbs and reduced in size distally; suture between first and second chin shields usually straight; second chin shields more than two thirds length of first chin shields (mean ratio 0.77, range 0.70–0.88); and adults with a plain grey to purplish brown dorsal colouration across the head and body with no or at most a faint pattern.

Further diagnosed from other species within the *G. australis* complex genetically by 10 unique amino acids in the *ND2* locus (Table 1).

*Gehyra lauta* sp. nov. overlaps geographically with three other species in the *G. australis* group: *G. dubia*, *G. gemina* sp. nov. and *G. robusta*. It differs from *G. dubia* by its larger size (mean and maximum adult SVL, respectively: 72.2 mm and 83.1 mm vs. 56.5 mm

and 64.7 mm), higher number of pre-cloacal pores (22–32 vs. 12–20), and its relatively plain grey dorsal colouration in life (vs. at least some darker spotting or mottling, and often also a distinct postorbital stripe); from *G. robusta* in having little or no dorsal pattern (vs. distinct brown and light grey spots and stripes) and higher number of pores in males (22–32 vs. 12–17); and from *G. gemina* sp. nov. by its larger size (mean and maximum adult SVL, respectively: 72.2 mm and 83.1 mm vs. 62.1 mm and 68.9 mm), higher number of pores (22–32 vs. 10–16), often extending onto limbs and reduced in size distally, posterior edge of first infralabial generally ~50% or greater the length of second supralabial (vs. ~60% or less) and outer edge of first pair of chin shields in contact with second pair usually strait, rarely slightly concave (vs. usually convex) and second pair of chin shields usually more than two-thirds length of first chin shields (mean and range ratios 0.77 (0.70–0.88) vs. 0.64 (0.56–0.75)).

Of the remaining species in the *G. australis* complex, *G. australis* and *G. arnhemica* sp. nov., the relatively disjunct and allopatric distribution permits identification in most areas where accurate locality data is available. Morphologically, *G. lauta* sp. nov. differs from *G. australis* in its larger size (mean and maximum adult SVL, respectively: 71.4 mm and 83.1 mm vs. 62.7 mm and 68.1 mm), second chin shields usually more than three-quarters length of first chin shields (mean and range ratios 0.77 (0.70–0.88) vs. 0.63 (0.58–0.70)), and more numerous pre-cloacal pores (22–32 vs. 10–16), usually extending onto limbs, with distal pores no greater than one-half the size of median pore (vs. not extending onto limbs, with distal most pores no smaller than one-half size of median pore). From *G. arnhemica* sp. nov., it differs by its larger size (mean and maximum adult SVL, respectively: 71.4 mm and 83.1 mm vs. 63.0 mm and 68.2 mm).

*Description*. As for *G. australis* complex description treated herein above, with the differences and variation outlined in the diagnoses above and in Table 2.

*Colour and pattern*. Photographs of the holotype (QM J90707), paratypes (QM J96621–4) and uncollected adults (Calvert River, NT, Australia) in life show adults to have a pale silvery-grey to purplish-brown dorsal surface with generally no discernible pattern, or at most indistinct lighter off-white regions on the body and tail (Figs. 8E and 8F). Juveniles (QM J96621 and CCM318 with SVL 42–43 mm) are more silvery brown with stronger, but still indistinct dark brown transverse bands or blotches. These differences in colouration suggest that this species undergoes an ontogenetic colour change, but colouration information from more samples is required to confirm this. In preservative, dorsal surfaces of head, body, limbs and tail varies from medium grey to light brownish grey to with no discernible pattern. Ventral surfaces buff, sometimes with fine greyish maculations along the lateral edges and/or forming indistinct mottling across subcaudal surfaces.

*Summary description of holotype (Q. J90707)*. All measurement in mm: SVL 72.7; TrunkL 36.8; TrunkW 13.1; ForelimbL 9.4; HindlimbL 11.2; HeadL 17.2; HeadD 8.4; HeadW 13.9; SnoutL 7.4; SnoutD 6.0; ToeL 6.9. Rostral flat and rectangular, with medial crease

on upper third; internasals 2; supralabials to mid-point of eye 9 on right and 8 on left. Infralabials 10 on both sides; mental scale pentagonal with rounded anterior edges; chin shields elongate, rounded and in three pairs, second pair more than two thirds length of first pair; parainfralabials rounded, heterogenous in size from small to moderately large. Body slightly compressed. Limbs relatively short, subapical subdigital lamellae undivided, 9 on fourth right toe. Tail regrown. Pores 26, largest median pores up to approximately 5 times diameter of smallest distal most pores.

*Distribution, habitat and ecology.* Restricted to the rocky ranges of north-western Qld and north-eastern NT, genetically verified records extend from China Wall in the west to Mount Gordon in the Selwyn Ranges to the east (Fig. 2). All records of *G. australis* from further east in Qld are likely *G. dubia* or different currently unrecognised *Gehyra* taxa (P. Couper, C. Hoskin, 2018, personal communication). While this species is generally known from areas with rocky ranges, within this habitat it is almost always collected from tree trunks, especially smooth barked *Corymbia* sp. or *Grevilla* sp. (P.M. Oliver, C. Moritz, 2010, personal observations), with *G. robusta* occurring on nearby rocky microhabitats.

*Etymology.* The species epithet is from the Latin word *lautus* (washed, neat, elegant), in reference to the plain or washed out dorsal pattern of the species.

## Systematics of the *Gehyra koira* complex
### Summary assessment of species diversity

Previous analyses of exonic sequences identified arboreal individuals resembling *G. australis* from the western Kimberley as being a distinct lineage within the otherwise saxicolous *G. koira* complex. *Gehyra chimera* sp. nov. is phylogenetically deeply nested within the *G. koira* complex, with different datasets variably grouping it with the western and/or southern Kimberley taxa *G. ipsa* and *G. calcitectus* sp. nov. (*Oliver et al., 2019*; Fig. 1). However, this taxon is ecologically more similar to the *G. australis* complex, being primarily arboreal and often found far from rocky microhabitats. Phenotypically it resembles members of the *G. australis* complex, especially *G. gemina* sp. nov. in overall morphology (Figs. 4A and 10), size and grey dorsal colourations with dark-brownish vermiculations. On the basis of morphological and genetic divergence from other members of the *G. koira* complex, we first present a description of this lineage as a new species.

Within the remaining largely saxicolous populations of the *G. koira* complex, prior phylogenetic and coalescent analyses of exon sequences (*Oliver et al., 2019*; Fig. 1) identified four candidate species—the large-bodied *G. ipsa* and three parapatric lineages within *G. koira* (s.l.). For these, geographically comprehensive sampling for nDNA SNPs revealed cohesive and, in PCoA, clearly separated genetic clusters, including individuals from parapatric boundaries. As for the *G. australis* complex, there was strong concordance of mtDNA clades with nDNA genetic identity. The statistical (conStruct) analysis yielded results less clear than was the case for the *G. australis* group, which we tentatively attribute to inability to model isolation by distance in taxa with restricted

(*G. ipsa*) or naturally disjunct (*G. calitectus* sp. nov.) distributions. In this group, there is clearer morphological separation of taxa (Fig. 4C), with consistent morphological differentiation in one or more of the following traits (with the outliers noted), overall size (*G. ipsa*), colour pattern (*G. lapistola* sp. nov., *G. calitectus* sp. nov.), pore number (*G. calitectus* sp. nov.) and aspects of scalation (*G. ipsa*). There was also some evidence of ecological differentiation or displacement with *G. calitectus* sp. nov., which is known only from widely disjunct patches of limestone karst (with *G. koira* on adjacent sandstones), suggesting a close association with this particular habitat. Combining evidence for genetic cohesion (PCoA), prior analyses of exonic sequence data and morphological differences, we recognise each of these four lineages as separate species.

On the basis of geography and morphology two of these lineages unequivocally correspond to named taxa; *G. ipsa* is a large bodied form with a restricted distribution on the Purnululu Massif in the eastern Kimberley, while the smaller and much more widespread koira 1 corresponds with true *G. koira* (type locality Keep River National Park, NT, Australia) (*Horner, 2005*). When originally described the distribution of these two taxa was widely disjunct; however, subsequent sampling has filled the gap, and genetic data show that both occur in the Purnululu Massif, posing challenges for identification of smaller and non-genotyped specimens from this area (see below under the redescription of *G. ipsa*).

No scientific names have previously been proposed for either the arboreal taxon in the western Kimberley, or the three other saxicolous lineages in the *G. koira* complex, so we formally name each of these herein.

### *Gehyra chimera* sp. nov.
urn:lsid:zoobank.org:act:E2BF5779-651D-4CB4-BF67-B7E286750DB2
Western Kimberley Tree Gehyra
koira4 of *Oliver et al. (2019)*
Figs. 6, 7, 10A, 10B, and 16

*Holotype*. WAM R177687 (field # CCM3372), adult male, collected from Bell Creek Crossing, Gibb River Road, WA (−17.1686°S, 125.3611°E), collected by P.M. Oliver, P. Skipwith and G. Armstrong, 9 November 2014.

*Paratypes* (N = 14*). Western Australia*: WAM R177673 (field # CCM0694), WAM R177674 (field # CCM0714), Doongan Station homestead, WA (−15.37866°S, 126.31163°E); WAM R177675 (field # CCM0873), Gibb River-Kalumburu Road, 25 km W Theda homestead, WA (−14.834°S, 126.3001°E); WAM R177682 (field # CCM1239), WAM R177683 (field # CCM1240), Silent Grove Ranger Station, WA (−17.0666°S, 125.2501°E); WAM R177684 (field # CCM1287), Mt Hart Station, WA (−16.8184°S, 124.9209°E); WAM R177685 (field # CCM3370), WAM R177686 (field # CCM3371), Bell Creek Crossing, Gibb River Road, WA (−17.1686°S, 125.3611°E); WAM R177688 (field # CCM3375), WAM R177689 (field # CCM3376), Silent Grove campground, Wilinggin Conservation Park, WA (−17.0669°S, 125.2476°E); NMV D76976, Silent Grove campground, Wilinggin Conservation Park, WA (−17.0677°S, 125.2477°E); NMV

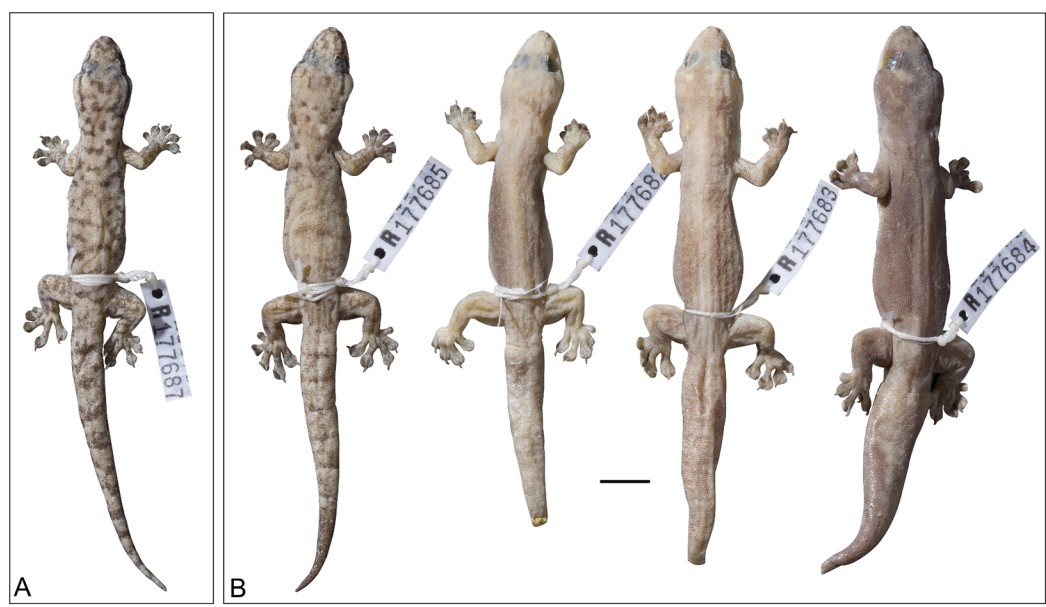

**Figure 16 Variation amongst preserved type series of *Gehyra chimera* sp. nov.** From left to right, (A) holotype (WAM R177687) and (B) four paratypes (WAM R177685, WAM R177682, WAM R177683, WAM R177684). Scale bar = 10 mm.

D77000, King Leopold Ranges, on Gibb River Road, WA (−17.1298°S, 125.2428°E); NMV D77043–44, Mt Hart Station, WA (−16.8180°S, 124.9207°E).

*Referred material.* See Tables S1–S2.

*Diagnosis.* A large *Gehyra* species (up to 73.7 mm SVL) most similar morphologically to members of the *G. australis* complex (despite occurring within the *G. koira* complex), differing from all other *Gehyra* species outside the *G. australis* complex (including other members of the *G. koira* complex) as per the group diagnosis above. Differs from members of the *G. australis* and *G. koira* complexes in the combination of: large size compared to *G. australis* complex members and moderate size within the *G. koira* complex (adult SVL up to 73.7 mm, mean 62.5 mm); in males 9–11 pre-cloacal pores in shallow chevron; suture between first and second chin shields usually straight; second chin shields usually just over two thirds length of first chin shields (mean ratio 0.75, range 0.68–0.80); usually 3–4 internasal scales, rarely 2; base colouration of dorsum in life usually greyish and either plain and without pattern, or at most with greyish brown vermiculations and scattered spots.

Further diagnosed from other members of the *G. australis* and *G. koira* complexes genetically by two unique amino acids in the *ND2* locus (Table 1). Some specimens, particularly juveniles and females, may only be diagnosable from of parapatric and potentially sympatric *G. gemina* sp. nov. and *G. koira* specimens on the basis of locality and/or genetic data.

*Gehyra chimera* sp. nov. is most similar to *G. gemina* sp. nov.; however, in addition to diagnostic genetic data in the ND2 gene, *G. chimera* sp. nov. also tends to differ in having:

longer second chin shields, usually just over two thirds length of first chin shields (mean ratio 0.75, range 0.68–0.80) vs. usually two thirds or less length (mean ratio 0.67, range 0.56–0.75); higher number of internasals (3–4 (80.6%), rarely 2 (19.4%) vs. usually 2 (61.9), rarely 3 (38.1%)); and a deeper snout (SnoutD/SVL mean ratio 0.84, range 0.81–0.87 vs. mean ratio 0.79, range 0.71–0.85).

*Gehyra chimera* sp. nov. abuts or overlaps with the distribution three other medium to large *Gehyra* in the western Kimberley (*G. koira*, *G. occidentalis* King and *G. xenopus* Storr), from which it differs as follows: from *G. koira* by its smaller size (mean and maximum adult SVL, respectively: 62.5 mm and 73.7 mm vs. 72.5 mm and 80.4 mm), lower number of pre-cloacal pores in males (9–11 vs. 13–23), and its relatively plain grey dorsal colouration in life (vs. brownish with at least some transverse barring or banding); from *G. occidentalis* in having undivided subdigital lamellae (vs. divided); and from *G. xenopus* in lacking wedge shaped patch that divides the proximate subdigital lamellae and in its greyish dorsal pattern (vs. mid-brown with numerous light and dark ocelli).

*Description*. As for *G. australis* complex description treated herein above, with the differences and variation outlined in the diagnoses above and in Table 2.

*Colour and pattern*. Based on photographs of the holotype and paratypes (WAM R177683–9) from the King Leopold Ranges, base colouration of all dorsal and lateral surfaces purplish-grey to very light pinkish grey, overlain by light to dark grey blotches, vermiculations, flecking and/or distinct bands, sometimes with additional very indistinct and thin light grey bands, and usually also with grey post- and pre-orbital stripes (Fig. 10). Venter mostly off-white with relatively little pattern. In preservative dorsal background colouration light brownish grey, with either no further pattern, or dark-brown vermiculations and blotching on the head and/or across the dorsum and tail. Ventral surfaces of body pale cream, with or without extensive fine grey maculations especially towards lateral edges and on throat. Subcaudal surfaces usually pale cream with extensive grey maculations laterally, and occasionally also forming thin and indistinct subcaudal bands.

*Summary description of holotype* (*WAM R177687*). All measurement in mm: SVL 56.1; TrunkL 23.4; TrunkW 9.2; ForelimbL 6.0; HindlimbL 7.1; HeadL 12.9; HeadD 6.1; HeadW 10.3; SnoutL 5.3; SnoutD 4.6; ToeL 5.5. Rostral notched dorsally, with medial crease in dorsal third. Nostrils separated by three internasal scales, with medial scale much smaller; supralabials to mid-point of eye 7 on both sides. Infralabials 8 on both sides; mental scale pentagonal; chin shields in two pairs, first pair elongate and second pair rounded; parainfralabials rounded, heterogenous in size. Limbs relatively short, with 8 undivided lamellae on right fourth toe. Tail original. Pre-cloacal pores 9, arranged in a broad chevron formation.

*Distribution, habitat and ecology*. Found in the west of the Kimberley region of WA, as far south as Bell Gorge in the King Leopold Ranges, west to Koolan and Kingfisher

Islands on the northern edge of the Yampi Peninsula, and north as far as Theda Station (Fig. 2). Almost exclusively known from trees in savannah woodlands, and also regularly found on human structures such as amenities blocks in the same habitat. It occurs in close proximity to, but not sympatric with, *G. gemina* sp. nov. in the King Leopold Ranges, Prince Regent River NP and at Theda Station.

*Etymology*. The species named after Chimera, a monstrous hybrid creature of Greek mythology composed of parts of multiple animals, pertaining to the close morphological similarity to the *G. australis* complex juxtaposed against clear genetic membership in the *G. koira* complex.

## *Gehyra koira* complex (excluding *G. chimera*): diagnosis and descriptions

A group of medium sized to large *Gehyra* (max SVL 77.1–94.9 mm), body shape slightly dorsoventrally compressed with fine homogenous rounded scales on dorsum and flattened scales on ventrum, snout moderately long with rounded tip, eyes large and protruding, ear opening small, large internasals bordering nares separated by 0–6 smaller internasals of greatly varying size, nostril in contact with rostral scale, limbs short with claws on digits II–V, claws protruding from dorsal surface of expanded toe pad, no skinfold behind the hindlimbs, subdigital lamellae under fourth toe undivided or at most with a weak medial groove, tail cylindrical tapering to a fine point, dorsum in life usually greyish to brown, and either plain and without pattern or more usually with some dark and light banding, vermiculations and scattered spots, in males 9–25 pre-cloacal pores arranged in a shallow chevron formation, and in females two eggs per clutch.

### *Gehyra koira* Horner, 2005
King's Rock Gehyra
koira1 of *Oliver et al. (2019)*
Figs. 6, 7, 9A, 9B and 17

*Holotype*. NTM R22406, adult female, collected from Nganlang Art Site, Keep River NP, NT (−16.0409°S, 128.9113°E), collected by P. Horner, 27 April 1996.

*Paratypes* (*N* = 45). *Western Australia*: NTM R7043, Kununurra, WA; NTM R7113–14, Kununurra, WA. *Northern Territory*: NTM R3875, Wickham River, Gregory NP, NT; NTM R9125–27, R9130, R9132, Keep River NP, NT; NTM R9471, Victoria River Bridge area, Victoria Highway, NT; NTM R10079, R10518–19, R10521–22, R10524–26, Keep River NP, NT; NTM R12770, Jasper Gorge, Buchanan Highway, NT; NTM R13265, Victoria River Bridge area, Victoria Highway, NT; NTM R18626–28, Bradshaw Station, NT; NTM R20774–75, Timber Creek, NT; NTM R22404, Jarrnarm Escarpment, Keep River NP, NT; NTM R22910, Spirit Hills, NT; NTM R23365, Jellabra Rockhole, Tanami Desert, NT; NTM R23754–55, Lonely Springs Creek Crossing, Buntine Highway, NT; NTM R23800, R23804, Wickham River, Gregory NP, NT; NTM R24103, North Kollendong Valley, Bradshaw Station, NT; NTM R24251–52, R24254–55, Fitzmaurice

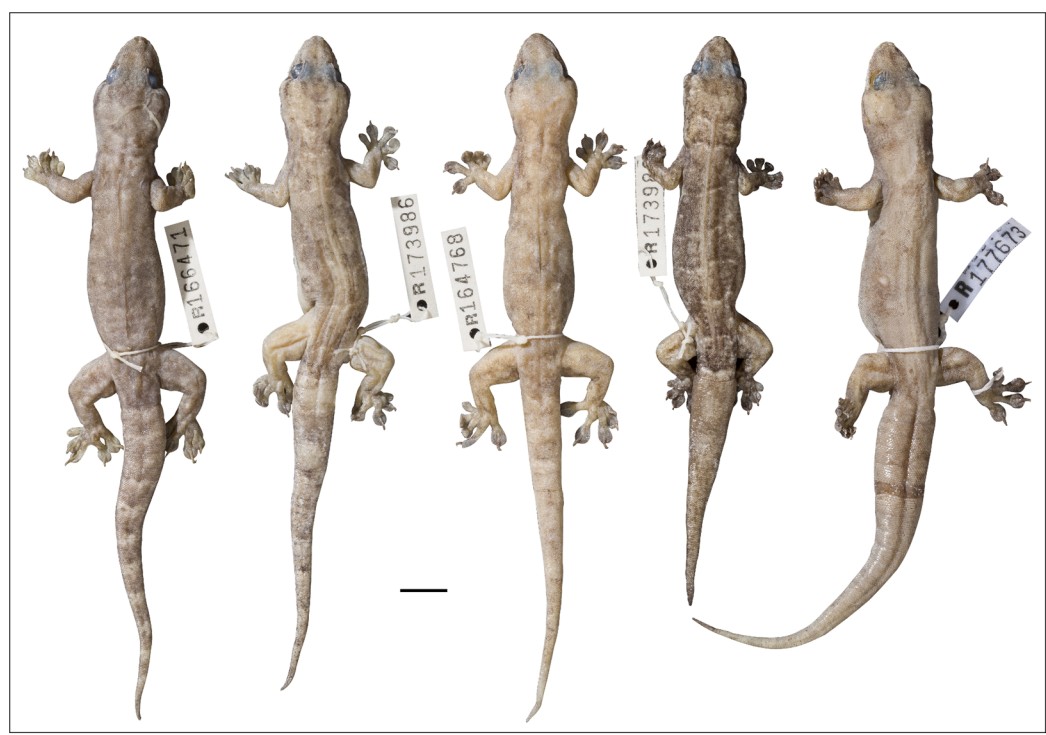

**Figure 17 Variation amongst preserved *Gehyra koira*.** From left to right, WAM R166471, WAM R173986, WAM R164768, WAM R173982, WAM R177673. Scale bar = 10 mm.

River, Bradshaw Station, NT; NTM R24623, Augustus Hole, Spirit Hills Station, NT; NTM R24850, R24853, Lobby Creek, Bradshaw Station, NT; NTM R25499, Jasper Gorge, Buchanan Highway, NT; NTM R27279, Pigeon Hole, Gregory NP, NT; WAM R108446 (formerly NTM R10520), Keep River, NP, NT; WAM R108447 (formerly NTM R22462), Limestone Gorge, Gregory NP, NT; WAM R108448 (formerly NTM R24253), Fitzmaurice River, Bradshaw Station, NT.

*Referred material.* See Tables S1 and S2.

*Diagnosis.* A large *Gehyra* species (up to 80.4 mm SVL), differing from all other *Gehyra* species outside of the *G. koira* complex as per the diagnosis above. Differs from other members of *G. koira* complex in the combination of: large size (adult SVL up to 80.4 mm, mean 72.5 mm); pre-cloacal pores in males numerous (12–23); second chin shields almost always more than two thirds length of first chin shields (mean ratio 0.72, range 0.64–0.79); first chin shield pair not bordered posteriorly by a single enlarged medial gular scale, or if present, median scale is not the largest in first row of gulars; adults in base colouration with tan to brown dorsum and tails, usually with some distinct transverse light and/or dark barring and a distinct to indistinct brown postorbital stripe.

Further diagnosed from other species within the *G. koira* complex genetically by one unique amino acid in the *ND2* locus (Table 1).

*Gehyra koira* occurs in sympatry or in close geographic proximity to six other species of moderate to large-sized *Gehyra*. Two of these are distantly related and can be easily distinguished: *G. koira* differs from *G. occidentalis* in having undivided lamellae (vs. divided); and from *G. xenopus* in lacking a wedge-shaped patch of granules that divide the proximal subdigital lamellae and dorsal pattern of barring or banding (vs. numerous light and dark ocelli). *Gehyra koira* can be distinguished from the four remaining sympatric taxa (all members of the *australis* group) as follows: from *G. ipsa* by its smaller size (mean and maximum adult SVL, respectively: 72.5 mm and 80.4 mm vs. 84.9 mm and 94.9 mm) and absence of an enlarged medial gular scale behind the first pair of chin shields, or when present, median scale not the largest in first row of gular scales (vs present and always the largest in first gular row); from *G. calcitectus* sp. nov. (see below) in having a dorsal pattern generally lacking light-coloured ocelli (vs. usually present); from *G. gemina* sp. nov. in its larger size (mean and maximum adult SVL, respectively: 72.5 mm and 80.4 mm vs. 62.9 mm and 68.9 mm) and higher number of pores (13–23 vs 10–16); and from *G. lapistola* sp. nov. (see below) by its brown dorsum with light transverse bars and or spots (vs. generally plainer and almost unpatterned) and generally higher number of pores (13–23 vs. 9–13).

*Description.* As for *G. koira* complex description treated herein above, with the differences and variation outlined in the diagnoses above and in Table 2.

*Colouration and pattern.* In life, base colouration of dorsal and upper lateral surfaces brownish-grey to medium-brown (Figs. 9A and 9B). Torso pattern variable, sometimes unpatterned, but usually with 5–8 very pale grey irregular transverse dorsal bands or series of blotches which themselves are sometimes bordered by patches of darker brown pigmentation, additional light yellowish-grey dorsal spots or blotches also often present. Head usually with faint to strong brown postorbital and canthal stripes, and often light yellowish spotting and flecking. Original tails usually with same base colouration as dorsum, and often also with light grey and dark brown banding. Regrown tails unpatterned. In preservative, dorsal ground colour grey to light brown, usually, but not always, with faint to distinct thin lighter grey and/or dark brown transverse banding or barring. Head same colour as torso, unpatterned, or with indistinct paler and/or darker brown spotting and blotching, dark postorbital stripe also apparent in approximately half of the recently preserved specimens examined. Ventral surfaces of torso, head, limbs and buff, largely unpatterned, but often with scattered greyish-brown maculations, especially on throat, limbs and latero-ventral surfaces of torso.

*Distribution, habitat and ecology.* Widespread across approximately the eastern half of the Kimberley region of WA, occurring as far west as Theda Station, north to the coast, including some offshore islands such as Adolphus and Sir Graham Moore Islands, and as far south as the limestone ranges on Mt Piere Station (Fig. 2). It also occurs widely in the escarpments of the Victoria River region of the western portion of the NT, extending at least to the eastern block of Gregory/Judbarra NP.

Occurs in most rocky range habitats within its distribution including granite, sandstone and limestone ranges. It is often observed on open rocky faces several metres from cover or retreats, and also forages on the trunk and branches of trees and shrubs within rocky areas. It overlaps with the ecologically similar *G. occidentalis* at some sites in the southern Kimberley, with known sympatry at one site at Mornington (C. Moritz, 2016, personal observations).

### *Gehyra ipsa* (Horner, 2005)

Bungle Bungle Ranges Gehyra
ipsa of *Oliver et al. (2019)*
Figs. 6, 7, 9C, 9D and 18

*Holotype*. WAM R101238, adult male, collected from Piccaninny Massif, Purnululu (Bungle Bungle) NP, WA (−17.45°S, 128.40°E), collected by N. Gambold, 25 August 1989.

*Paratypes (N = 11)*. *Western Australia*: NTM R28098 (formerly WAM R104285), Echidna Chasm, Purnululu (Bungle Bungle) NP, WA; NTM R28099 (formerly WAM R101239), Picaninny Massif, Purnululu (Bungle Bungle) NP, WA; WAM R101237*, Picaninny Massif, Purnululu (Bungle Bungle) NP, WA; WAM R104283*, Osmand Valley Station, WA; WAM R104284, Frog Hole Gorge, Purnululu (Bungle Bungle) NP, WA; WAM R104285, R104290–91, Echidna Chasm, Purnululu (Bungle Bungle) NP, WA; WAM R104286, R104288, R104295, Picaninny Gorge, Purnululu (Bungle Bungle) NP, WA; WAM R104289*, Wulwuldji Spring, Purnululu (Bungle Bungle) NP, WA.

Three paratypes marked with an asterisk are treated herein as *G. koira* and not true *G. ipsa* based on combination of medial scale of first gular row not being enlarged, or if enlarged, not the largest, small size (<75 mm SVL), and co-occurrence with specimens genotyped as *G. koira*. In the absence of dorsal pattern lacking distinct light tan ocelli or blotches, specimens were not considered to be assignable to *G. calcitectus* sp. nov. Due to the morphological overlap and potential occurrence of sympatry of both species, in addition to *G. calcitectus* sp. nov., there remains some uncertainty as to the species in which these specimens apply.

*Referred material*. See Tables S1 and S2.

*Diagnosis*. A large *Gehyra* species (up to 94.9 mm SVL), differing from all other *Gehyra* species outside of the *G. koira* complex as per the diagnosis above. Differs from other members of the *G. koira* complex in the combination of: large size within complex (adult SVL up to 94.9 mm, mean 84.9 mm, largest of the *G. koira* complex); pores in males moderately numerous (15–18); first chin shield pair bordered posteriorly by a single enlarged medial scale, ~1.5–4 times the size of adjacent scales and always the largest in first row of gular scales posterior to chin shields; second chin shields approximately two thirds length of first chin shields (mean ratio 0.78, range 0.75–0.81); adults in base colouration with tan to brownish dorsum with light transverse barring, a distinct to indistinct brown postorbital stripe, tails usually with at least some distinct transverse light and/or dark barring.

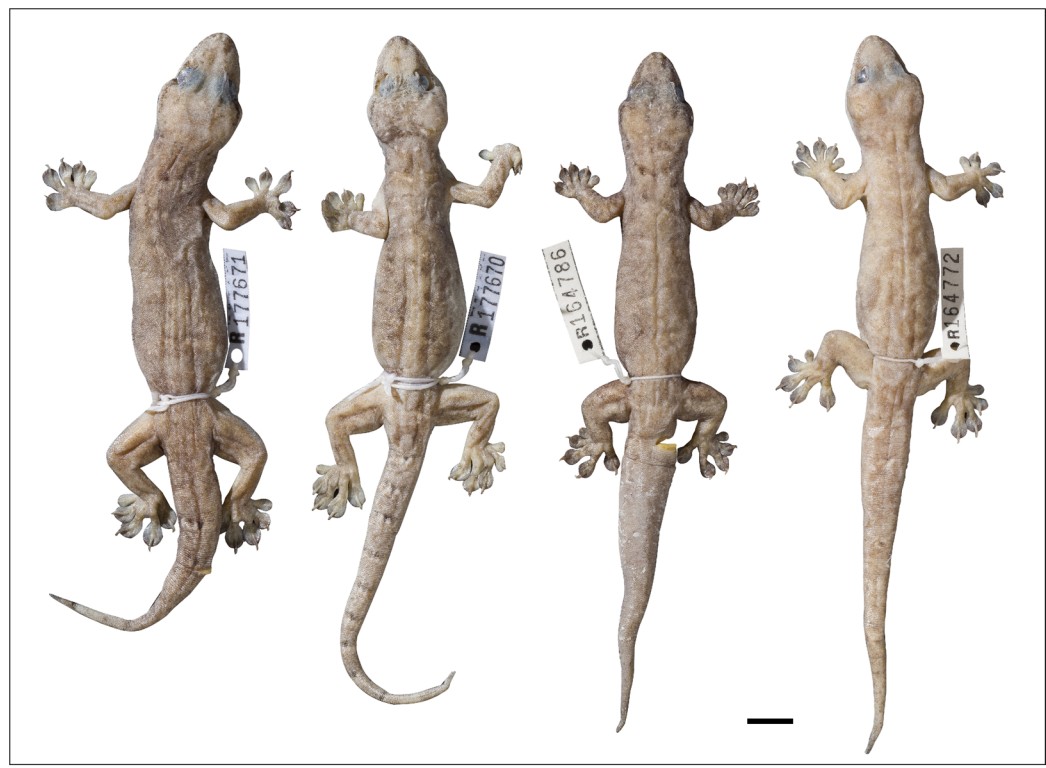

**Figure 18 Variation amongst preserved *Gehyra ipsa*.** From left to right, WAM R177671, WAM R177670, WAM R164786, R164772. Scale bar = 10 mm.

Further diagnosed from other species within the *G. koira* complex genetically by four unique amino acids in the *ND2* locus (Table 1).

*Gehyra ipsa* is likely to occur in parapatry or even sympatry with *G. koira* (which it is morphologically most similar to), *G. gemina* sp. nov. and possibly *G. calcitectus* sp. nov. It can often be differentiated from all three by the combination of larger size (mean and maximum adult SVL, respectively: 84.9 mm and 94.9 mm), presence of an enlarged medial scale behind the first pair of chin shields, which is always the largest in the first row of gular scales posterior to chin shields (vs. absence or, if present, not the largest scales in first gular row) and dorsal pattern comprising transverse light bands (vs. often comprising light pale tan ocelli or blotches in *G. calcitectus* sp. nov.).

*Description*. As for *G. koira* complex description treated herein above, with the differences and variation outlined in the diagnoses above and in Table 2.

*Colour and pattern*. In life base colouration of dorsum, head, limbs and tail light to medium-brown (Figs. 9C and 9D). Patterned with light grey barring on the torso, flecking on the posterior portion of head, and barring on the tail, and further dark brown postorbital stripes and jagged tail bands. Original tails usually same base colouration as dorsum, sometimes with light-grey and/or dark brown banding. Regrown tails unpatterned. In preservative, dorsal ground colour light greyish-brown, often with faint to distinct lighter grey and/or dark-brown transverse banding or barring. Head occasionally with further light

grey spotting, and/or an indistinct medium brown postorbital stripe. Ventral surfaces of torso, head, limbs and buff, largely unpatterned, but often with scattered greyish-brown maculations, especially on throat, limbs and latero-ventral surfaces of torso.

*Distribution, habitat and ecology. Gehyra ipsa* is known only from the western edge of the Bungle Bungle Range in Purnululu NP, located in the south-west of the Kimberley region of WA (Fig. 2). It seems to be more abundant in the southern-western area spanning Piccaninny Gorge, out along Piccaninny Creek and Whipsnake Gorge. Previous records previously assigned to this species from along the northern edge of the massif and nearby sites are considered to be *G. koira. Gehyra ipsa* is usually observed on open rock faces (especially within gorges) and often occurs in close association with small *Ficus* trees growing from the rock faces.

*Remarks.* Further detailed geographic and genetic sampling within the Bungle Bungle Range, particularly along the northern edge, is required to better understand patterns of geographic, ecological and genetic interaction between *G. ipsa* and *G. koira*.

Examination of the type series of *G. ipsa*, including the 11 putative *G. ipsa* paratypes, revealed three of the 11 paratypes lacked the diagnostics corresponding with the species treatment herein. The three specimens lacked an enlarged median gular scale posterior to first pair of chin shields that was the largest in the first row of gular scales (Fig. 7) and were also substantially smaller than the maximum size reached by the type specimen (WAM R101238) and genotyped *G. ipsa*. Where an enlarged median gular scale was present in genotyped *G. koira*, the scale was generally not the largest present in the first row of gular scales posterior to the chin shields. On this basis we propose the hypotheses that: (i) an enlarged medial scale behind the first pair of chin shields is diagnostic for *G. ipsa*, (ii) *G. ipsa* is restricted to the southern edge of the Purnululu Massif and (iii) *G. ipsa* and *G. koira* occur in sympatry on the southern edge of the Purnululu Massif. Therefore, based on present data available, we redefine the distribution of both species, and treat some paratypes of *G. ipsa* as *G. koira* (see paratypes above).

The formal elevation and redefinition of its range of *G. koira* brings the number of recognised short-range endemic lizards from the Bungle Bungle Ranges to two (the other species being *Lerista bunglebungle* Storr), with additional undescribed taxa known to occur within the ranges (B. Maryan, 2018, personal communication). This suggests that the distinctive geology and deep gorges of the Bungle Bungle Range has mediated localised persistence and divergence, similar to nearby limestone ranges (*Oliver et al., 2017*).

### *Gehyra lapistola* sp. nov.
urn:lsid:zoobank.org:act:37EA721F-35D0-4F22-A5FF-24B63EB2F91C
Litchfield Rock Gehyra
koira2 of *Oliver et al. (2019)*
Figs. 6, 7, 9E, 9F and 19

*Holotype.* NTM R37093, adult male, collected from Dorat Road, approximately 6 km from Stuart Highway, NT (−13.2851°S, 131.1174°E), collect by P.M. Oliver, M. Hammer and P. Skipwith, 28 September 2013.

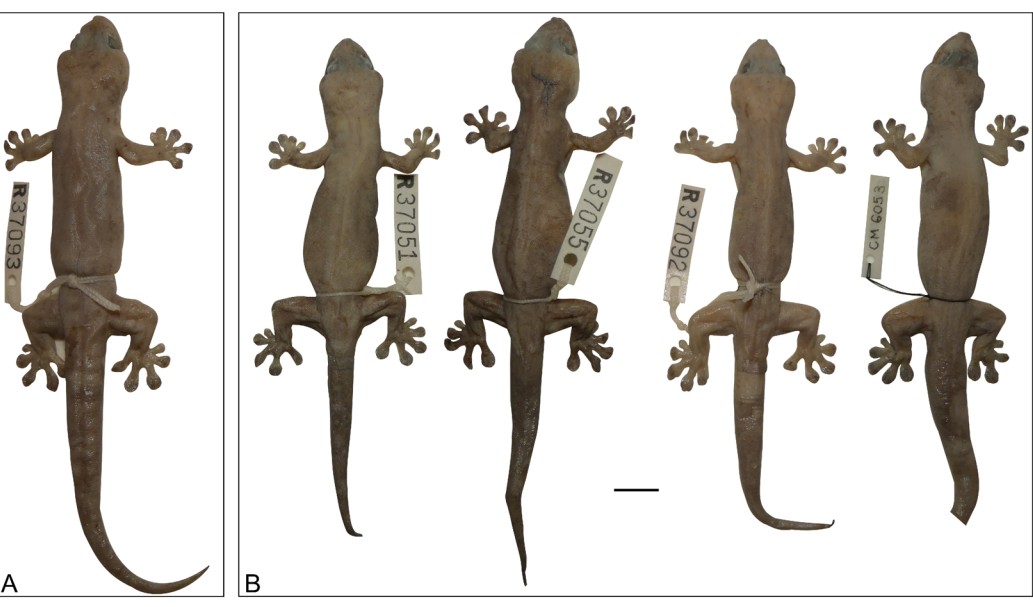

**Figure 19 Variation amongst preserved type series of *Gehyra lapistola* sp. nov.** From left to right, (A) holotype (NTM R37093) and (B) four paratypes (NTM R37051, NTM R37055, NTM R37092, NTM R38195). Scale bar = 10 mm.               

*Paratypes (N = 11). Northern* Territory: NTM R21786, Litchfield NP, NT (−13.22°S, 130.73°E); NTM R36708, Fish River Gorge, Fish River Station, NT (−14.2660°S, 130.8936°E); NTM R37055, Dorat Road, Robin Falls area, NT (−13.35°S, 131.14°E); NTM R37051, Daly River Road, Daly River region, NT (exact latitude and longitude unknown); NTM R37092, Dorat Road, approximately 6 km from Stuart Highway, NT (−13.2851°S, 131.1174°E); NTM R38191 (field # CCM2882), NTM R38192 (field # CCM2884), Tolmer Falls turnoff, Litchfield NP, NT (−13.1965°S, 130.7139°E); NTM R38193 (field # CCM2935), Florence Falls turnoff, Litchfield NP, NT (−13.1264°S, 130.8046°E); NTM R38194 (field # CCM6052), NTM R38195 (field # CCM6053), Mount Pleasant, Tipperary Station, NT (−13.5915°S, 131.1678°E); NTM R38196 (field # CCM6070), Litchfield Mining Camp, 5 km from Daly River Road, NT (−13.5096°S, 130.7727°E).

*Referred material.* See Tables S1 and S2.

*Diagnoses.* A large *Gehyra* species (up to 79.5 mm SVL), differing from all other *Gehyra* species outside the *G. koira* complex as per the diagnosis above. Differs from other members of the *G. koira* complex in the combination of: moderate size within complex (max SVL 79.5 mm, mean 74.2 mm); pre-cloacal pores in males not numerous (9–13); first chin shield pair not bordered posteriorly by a single enlarged medial gular scale, or if present, median scale is not the largest in first row of gulars; second chin shields approximately two thirds length of first chin shields (mean ratio 0.68, range 0.62–0.74); usually only two internasals (80% individuals); and adults with plain grey to brownish dorsum across the head and body with no distinct pattern of barring, banding or spots.

Further diagnosed from other species within the *G. koira* complex genetically by nine unique amino acids in the *ND2* locus (Table 1).

*Gehyra lapistola* sp. nov. is morphologically similar to the other geographically disjunct rock-dwelling members of the *G. koira* complex (*G. koira*, *G. ipsa* and *G. calcitectus* sp. nov.) that occur further to the west; it differs, however, in its plainer and often almost unpatterned dorsum (vs. usually brown with light transverse bars and/or blotches). It further differs from *G. koira* in generally having fewer pre-clocal pores (9–13 vs. 13–23) and a lower number of internasals (usually two (80%), rarely 3–4 (20%) vs rarely two (12.5%), usually 3–4 (87.5%)). From *G. ipsa*, it differs in fewer pre-cloacal pores (9–13 vs. 14–18) and first chin shield pair not bordered posteriorly by a single enlarged medial gular scale, or if present, median scale is not the largest in first row of gulars (vs. present and always largest scale of first gular row). From *G. calcitectus* sp. nov., it differs in having a dorsal pattern including pale transverse stripes (vs. light pale tan ocelli or blotches). Although geographically disjunct from other members of the *G. koira* complex, some specimens, especially juveniles, weakly patterned females or preserved specimens, may only be accurately diagnosed on the basis of locality and/or genetic data.

*Gehyra lapistola* sp. nov. is morphologically similar and overlaps geographically with one species of the *G. australis* complex, *G. australis*. From *G. australis*, it differs in its larger size (mean and maximum adult SVL, respectively: 74.2 mm and 79.5 mm vs. 62.7 mm and 68.1 mm) and more pale and plainer colouration in both preservative and life with little or no pattern (vs. usually with at least some dorsal pattern of darker brown spots or flecks). Comparison of preserved animals also suggests that the pore series is more sharply angled in *G. lapistola* sp. nov. than in *G. australis*; however, this character is difficult to accurately measure as it varies with angle of limb preservation.

*Description.* As for *G. koira* complex description treated herein above, with the differences and variation outlined in the diagnoses above and in Table 2.

*Colour and pattern.* Photos of live or recently euthanized adults shows fleshy pink, light cream or light brownish dorsal colouration (Figs. 9E and 9F). Head and tail have the same colouration as the body. In preservative, dorsal surfaces of head, body, limbs and tail grey to light brownish grey with no discernable pattern. Ventral surfaces of torso, head, limbs and tail off-white, usually unpatterned, but more rarely with scattered pale brownish-grey maculations, especially on throat, limbs and latero-ventral surfaces of torso.

*Summary description of holotype (NTM R37093).* All measurement in mm: SVL 79.5; TrunkL 34.2; TrunkW 12.5; ForelimbL 8.4; HindlimbL 10.0; HeadL 17.8; HeadD 8.9; HeadW 14.8; SnoutL 6.9; SnoutD 6.0; ToeL 7.5. A moderately large *Gehyra* (SVL 78.6 mm), head slightly narrow (HW/HL 0.77) and deep (HD/HL 0.96). Dorsal head scales small and granular, interorbitals 36; rostral protruding, with medial crease in upper third. Nostrils separated by two large internasal scales and one smaller intervening internasals scale with two granules above it; supralabials to mid-point of eye 10 on right side and 9 on left side. Infralabials 8 on both sides; mental scale with lunate posterior edge; chin shields elongate, rounded and in two enlarged and rounded pairs, second pair

approximately two thirds length of first; parainfralabials rounded, first approximately half diameter of second chinshield. Limbs relatively short, with 10 undivided lamellae on right fourth toe. No claw on fifth toes. Original tail intact, curved at tip. Pre-cloacal pores 13, arranged in a broad chevron formation.

*Distribution, habitat and ecology*. Restricted to the rocky ranges of north-eastern NT, especially in escarpments and outcrops in Litchfield NP, and extending south at least as far as Fish River Gorge on the other side of the Daly River (Fig. 2). *Gehyra lapistola* sp. nov. is usually collected on rocky substrate including both open rock faces and boulders, or more rarely on trees within this habitat.

    *Gehyra lapistola* sp. nov. has been observed in sympatry with three other *Gehyra* species: *G. australis*, *G. nana* and *G. paranana*. Where these distributions overlap they show evidence of ecological segregation, *G. lapistola* sp. nov. uses large rocks and open rock faces, *G. paranana* occurs on smaller rocks and boulders and closer to retreats, *G. nana* around the bottom of rock boulders and *G. australis* appears to be largely restricted to trees (C. Moritz, P.M. Oliver, 2013–2015, personal observations).

*Etymology*. The species epithet is formed from the Latin words *lapis* (rock, stone) and *stolo* (runner), used in its adjectival form as *stola*, as in 'rock-running', in reference to the species occurrence in rocky escarpment and outcrop habitats.

### *Gehyra calcitectus* sp. nov.
urn:lsid:zoobank.org:act:52919622-D6CD-4023-80B5-3ABEF515DB29
Relictual Karst Gehyra
koira3 of *Oliver et al. (2019)*
Figs. 6, 7, 9G, 9H and 20

*Holotype*. WAM R177691 (field # CCM3235), adult male, collected from Limestone Billy Hills, Gogo Station, WA (−18.3272°S, 125.7650°E), collected by P.M. Oliver, P. Skipwith and G. Armstrong, 3 November 2014.

*Paratypes* (N = 12). *Western Australia*. WAM R177692 (field # CCM3236), Limestone Billy Hills, Gogo Station, WA (−18.3272°S, 125.7650°E); WAM R177693 (field # CCM3259), south entrance of Menyous Gap, Pillara Range, Gogo Station, WA (−18.4044°S, 125.8370°E); WAM R177695 (field # CCM3333), WAM R177696 (field # CCM3334), Virgin Hills, near Bobs Bore, Gogo Station, WA (−18.5149°S, 125.9256°E); WAM R177697 (field # CCM7276), WAM R177698 (field # CCM7277), WAM R177699 (field # CCM7278), WAM R177700 (field # CCM7279, Lissadell Station, WA (−16.6635°S, 128.5265°E); WAM R177701 (field # CCM7330), Pillara Range, Gogo Station, WA (−18.3697°S, 125.7363°E); WAM R177702 (field # CCM7466), WAM R177703 (field # CCM7470), WAM R177704 (field # CCM7473), Duncan Road, Argyle Station, WA (−16.5964°S,128.9536°E).

*Referred material*. See Tables S1 and S2.

**Figure 20 Variation amongst preserved type series of *Gehyra calcitectus* sp. nov.** From left to right, (A) holotype (WAM R177691) and (B) four paratypes (WAM R177701, WAM R177704, WAM R177699, WAM R177697). Scale bar = 10 mm.

*Diagnosis*. A large *Gehyra* species (to 77.1 mm SVL) differing from all other *Gehyra* species outside the *G. koira* complex as per the diagnosis above. Differs from other members of the *G. koira* complex in the combination of: moderate size within complex (adult SVL up to 77.1 mm, mean 69.4 mm); head relatively wide (HW/SVL 0.19–0.21); nostrils usually separated by more than 2 internasal scales (87.5%), rarely 2 (14.3%); second chin shields approximately two-thirds the length of first chin shields (mean ratio 0.66, range 0.53–0.79); first chin shield pair not bordered posteriorly by a single enlarged medial gular scale, or if present, median scale is not the largest in first row of gulars; pores in males relatively few (9–14); and adults with tan to brownish dorsum and tails, and with a dorsal pattern including distinct light tan ocelli or blotches (as opposed to transverse light bands).

Further diagnosed from other species within the *G. koira* complex genetically by six unique amino acids in the *ND2* locus (Table 1).

*Gehyra calcitectus* sp. nov. differs from two members of *G. koira* complex with which its distribution abuts or overlaps as follows: from *G. koira* in having a dorsal pattern including light pale tan ocelli or blotches (vs. transverse bars), lower number of pre-cloacal pores (9–14 vs. 12–23), and for eastern populations in having dark-brown dorsal spots across the head and often torso (vs. absent); and from *G. ipsa* in its smaller size (mean and maximum adult SVL respectively 69.4 mm and 77.1 mm vs. 84.9 mm and 94.9 mm), absence of an enlarged scale behind the first pair of chin shields (vs. presence) and pale dorsal pattern elements consisting of ocelli or pale blotches (vs. pale transverse bands). From the geographically disjunct *G. lapistola* sp. nov., it differs in having a dorsal pattern comprising pale ocelli or blotches (vs. at most pale transverse bands).

The distribution of *G. calcitectus* sp. nov. is largely geographically disjunct from most members of the *G. koira* complex, including *G. ipsa*, *G. lapistola* sp. nov. and *G. chimera* sp.

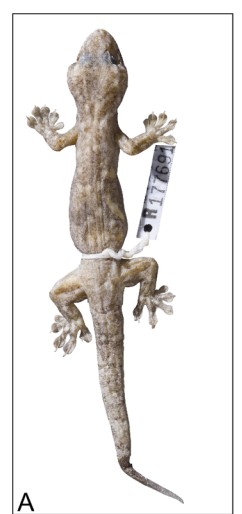

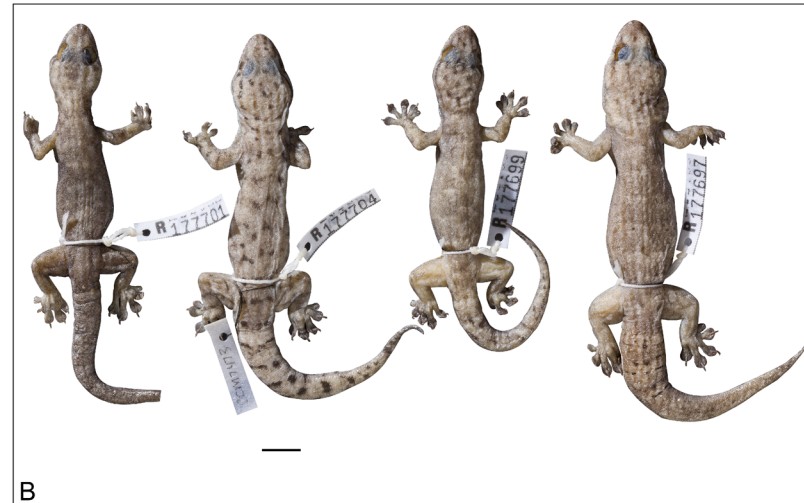

nov., and accurate locality data may permit identification of morphologically similar specimens.

*Description*. As for *G. koira* complex description treated herein above, with the differences and variation outlined in the diagnoses above and in Table 2.

*Colour and pattern*. In life, dorsal base colouration light to dark brown, always with rows of large pale yellowish-brown or off-white spots or ocelli across the back and usually head. Specimens from Lissadell Station and Argyle Rocks (Fig. 9G) in the east also tend to have dark brown spotting, blotching or striping on the head, and often torso, while specimens from Gogo Station (Fig. 9H) lack dark dorsal pattern. Limbs same base colouration as torso, and also usually with at least some pale brown blotching. Tail colouration usually consisting of bands or transverse series of blotches of varying in colour and width from thin pale brown bands on a medium brown background, to alternating broad dark brown and light brown bands. In preservative, dorsal surfaces light grey-brown, with weakly defined very pale grey ocelli usually, but not always, visible on body, limbs, and tail. Specimens from the two northern localities usually have a dark brown canthal stripe with further dark brown spots on the head, occasionally extending on to the torso and tail. Ventral surfaces of torso, head, limbs and tail pale cream, largely unpatterned, but often with scattered brownish-grey maculations, especially on throat, limbs and latero-ventral surfaces of torso and tail.

*Summary description of holotype* (*WAM R177691*). All measurement in mm: SVL 67.6; TrunkL 29.7 mm; TrunkW 11.2; ForelimbL 8.4; HindlimbL 8.7; HeadL 16.3; HeadD 8.3; HeadW 14.0; SnoutL 6.1; SnoutD 5.9; ToeL 6.9. Rostral flat, with medial crease in dorsal third. Nostrils separated by two large internasal scales; supralabials to mid-point of eye nine on both sides. Infralabials eight on right, seven on left; mental scale pentagonal; chin shields relatively short, rounded and in two enlarged and rounded pairs, second pair approximately two thirds length of first; parainfralabials rounded, first approximately half the width and height of second chinshield. Limbs relatively short, with nine undivided lamellae on right fourth toe. Tail largely original, with 20 mm regrown section at tip. Pre-cloacal pores 11, in a broad chevron formation.

*Distribution, habitat and ecology*. Known only from three isolated and disjunct limestone ranges along the southern and western edge of the Kimberley region, in Pillara Range on Gogo Station, and on Lissadell and Argyle Stations (Fig. 2).

At Argyle Station, WA, *G. calcitectus* sp. nov. were observed on low-lying pavement limestones and small faces (<3 m). At Gogo and Lissadell Stations they were observed on larger (>5 m) limestone faces and boulders. At Lissadell Station nearby sandstone ranges were occupied by *G. koira*. Similarily, only *G. koira* was found in limestone system in the east (Ningbing) and south (Mt Piere Station) Kimberley and in the Victoria Rivers District (Judburra NP).

*Etymology*. The species epithet is formed from the Latin words *calcis* (limestone) and *tectus* (hidden, hideaway), as in 'limestone-hidden' or 'limestone hideaway', in reference to the

species occurrence in and apparent preference for limestone habitats of the Kimberley limestone ranges. Used as a noun in apposition.

*Remarks. Gehyra calcitectus* sp. nov. is the fourth recently described or redescribed lizard species with a restricted range in the limestone ranges along the southern and eastern fringes of the Kimberley (*Oliver et al., 2014*, *2016a*; *Doughty, Ellis & Oliver, 2016*). Additional limestone endemics lineages from the southern and eastern Kimberley, and the Victoria Rivers District district are likely to represent additional undescribed species (C. Moritz, 2018, personal observations), further emphasising the biological significance and importance of limestone ranges in north-west Australia as hotspots of endemism and evolutionary refugia (*Oliver et al., 2017*; *Rosauer et al., 2018*).

## DISCUSSION

Documenting and describing morphologically similar, yet genetically and evolutionarily distinctive cryptic species poses well-recognised, but ongoing challenges for systematists, evolutionary biologists and conservation managers (*Bickford et al., 2007*; *Oliver, Keogh & Moritz, 2015*; *Singhal et al., 2018*). The existence of multiple species within the *G. australis* group was not apparent based on prima-facie morphological data. Indeed, in light of high levels of morphological similarity among many species, and given sparse geographic sampling in previous exon multilocus studies (*Noble et al., 2018*; *Oliver et al., 2019*), it was only after generating a SNP data set based on geographically extensive sampling that the case for the recognition of multiple evolutionarily distinct and isolated lineages (i.e. species) became compelling. The SNP data provided a crucial test for the lack of gene flow between closely related taxa in areas of geographic contact or even overlap. Many analyses of mtDNA diversity in other Australian lizard species have revealed similarly deep genetic divergences (*Oliver, Doughty & Palmer, 2012*; *Laver et al., 2017*; *Laver, Doughty & Oliver, 2018*), but corroborating independent evidence for evolutionary distinctiveness and the absence of contemporary of gene flow has been lacking. Where geographic sampling is sufficiently extensive, SNP data offer the potential to provide for direct tests of lack of recent genetic introgression between such problematic populations (*Singhal et al., 2018*). Conversely, increasing use of SNP methods may also serve to highlight instances of taxonomic oversplitting stemming from over-interpretation of morphological variation or results from genetic studies with inadequate sampling of geography and genes (*Georges et al., 2018*; *Hillis, 2019*).

Resolving species boundaries within morphologically conservative groups can yield new insights into speciation processes and eco-evolutionary drivers of spatial patterns of diversity (*Fišer, Robinson & Malard, 2018*). As cases in point, using genetic data to resolve species limits in *Gehyra* has revealed instances of parallel evolution (e.g. independently arboreal *G. chimera* sp. nov. and *G. australis* group; *Oliver et al., 2019*), body size evolution associated with establishment of sympatric assemblages (*Doughty et al., 2018a*; *Moritz et al., 2018*) and the high prevalence of short-range taxa in complex rocky environments (*Ashman et al., 2018*), and association of chromosome change with

speciation (*King, 1979*, *1983b*; *Moritz, 1986*). On the latter, *King (1983b)* karyotyped several individuals within *G. australis* s.l., all with the same 2N = 40a karyotype (but variable sex chromosomes) that by location can be assigned to *G. australis* s.s., *G. gemina* sp. nov. and *G. arnhemica* sp. nov. He also observed a different, 2n = 44 karyotype from several rock-dwelling individuals from central Kimberley which were ascribed to *G. occidentalis* by King (1984). There is no further information on chromosome variation across the taxa considered here, and given the association of such with speciation in *Gehyra*, especially in rock-dwelling forms, this should be rectified.

Many of the taxa described or revised here have wide geographic ranges and have been recorded from buildings and other anthropogenic structures, specifically: *G. australis*, *G. gemina* sp. nov. and *G. chimera* sp. nov. Our genetic data suggest that colonising of anthropogenic habitats has also led to human-mediated dispersal within northern Australia in some of these species. An isolated record of *G. gemina* sp. nov. from a roadhouse on the Stuart Highway in the NT at Daly Waters (SAMA R34176) is near genetically identical (mtDNA) to a clade within this species otherwise mainly known from the Kimberley region from WA (~500 km distant). More interestingly, the SNP dataset highlighted a single putative hybrid between *G. australis* and *G. gemina* sp. nov. that was collected on the Renner Springs roadhouse, an important stopover point for traffic along the main route (the Stuart Highway) running south of Darwin. This single anomalous individual from a highly disturbed anthropogenic setting raises the possibility that hybridisation has occurred as a result of human-assisted translocation and mixing of (potentially naïve) individuals of cryptic species.

In contrast, our partition of the mainly rock associated species in the *G. koira* complex has provided support for three taxa with relatively restricted ranges; that is qualifying as short-range endemics sensu *Harvey et al. (2011)*. Our revised taxonomy across the two complexes also highlights emerging biodiversity hotspots across northern Australia, including the Selwyn Range (*Oliver & Doughty, 2016*; *Noble et al., 2018*), Litchfield escarpment and surrounding areas (*Rosauer et al., 2016*), the Purnululu region in the eastern Kimberley and the fringing limestone of the Kimberley Craton (*Oliver et al., 2017*; *Rosauer et al., 2018*). Most taxa in these hotspots are ecologically associated with rocks, although *G. lauta* sp. nov. is mostly arboreal, suggesting persistence on trees within the Selwyn Range refugia (*Noble et al., 2018*). Perhaps the taxon with the most enigmatic distribution, however, is the widely disjunct and morphologically variable *G. calcitectus* sp. nov. Known localities for this taxon indicate it is closely associated with limestone (suggesting ecological specialisation), does not occur in sympatry with widespread *G. koira* (suggesting displacement or competition) and occurs in widely disjunct localities (suggesting range contraction). One possible hypothesis is that populations of this species have been isolated by the expansion of sandy deserts over formerly more widespread limestone ranges along western and southern edge of the Kimberley Craton.

More generally, here, as in other systems (*Bickford et al., 2007*), cryptic species are emerging as a substantial component of diversity. For the AMT specifically, most low vagility vertebrate species that have been adequately surveyed genetically, even if just with a
few loci, have revealed deep and geographically fine-grained phylogeographic structure (*Oliver, Doughty & Palmer, 2012*; *Moritz et al., 2016*, *2018*; *Potter et al., 2016*; *Laver et al., 2017*; *Laver, Doughty & Oliver, 2018*). With a few exceptions (*Doughty, Ellis & Oliver, 2016*; *Afonso Silva et al., 2017*; *Moritz et al., 2018*), the taxonomic status of much of this diversity remains to be evaluated. But even with the conservative criteria used here, it is likely that the current taxonomy substantially underestimates the true species diversity of the AMT. However, even though there is often a delay between the publication of genetic data and the eventual taxonomic revisions, it is possible to incorporate the new phylogeographic knowledge into conservation assessments (*Coates, Byrne & Moritz, 2018*). For the AMT, this approach has revealed novel hotspots of diversity, with high conservation importance (*Rosauer et al., 2016*, *2018*; *Oliver et al., 2017*).

## ABBREVIATIONS

| | |
|---|---|
| **ANU** | Australian National University |
| **ANWC** | Australian National Wildlife Collection, Canberra, ACT |
| **NHMUK** | Natural History Museum of the United Kingdom, London, United Kingdom (formerly British Museum of Natural History) |
| **NMV** | Museums Victoria, Melbourne, Victoria |
| **NP** | National Park |
| **NT** | Northern Territory |
| **NTM** | Museum and Art Gallery of the Northern Territory, Darwin, NT |
| **Qld** | Queensland |
| **QM** | Queensland Museum, Brisbane, Qld. |
| **WA** | Western Australia |
| **WAM** | Western Australian Museum, Welshpool, WA |

## ACKNOWLEDGEMENTS

We thank the following people for providing access to specimens and samples in their care: Andrew Amey (QM), Gavin Dally (NTM), Conrad Hoskin (James Cook University), Eric Rittmeyer (Rutgers University), and Katie Date and Jane Melville (NMV). Chris Jolly, Steve Richards, Peter Waddington and Stephen Zozaya provided photographs. Numerous members of the Moritz Lab (ANU) in addition to Graham Armstrong, Philip Skipwith and Mike Hammer assisted with fieldwork and logistics. We also thank landholders, Parks staff, and Indigenous Rangers and Traditional Owners for access to their lands and for participating in fieldwork.

### Funding

This research was supported by grants from from the Australian Research Council to Paul Oliver (ARC LP120100081 and ARC DE140100220) and Craig Moritz (ARC

FL110100104). The funders had no role in study design, data collection and analysis, decision to publish, or preparation of the manuscript.

## Grant Disclosures
The following grant information was disclosed by the authors:
Australian Research Council: LP120100081, DE140100220 and FL110100104.

## Competing Interests
The authors declare that they have no competing interests.

## Author Contributions
- Paul M. Oliver conceived and designed the experiments, analysed the data, contributed reagents/materials/analysis tools, prepared figures and/or tables, authored or reviewed drafts of the paper, approved the final draft.
- Audrey Miranda Prasetya performed the experiments, analysed the data, prepared figures and/or tables, authored or reviewed drafts of the paper, approved the final draft.
- Leonardo G. Tedeschi performed the experiments, analysed the data, prepared figures and/or tables, approved the final draft.
- Jessica Fenker performed the experiments, analysed the data, prepared figures and/or tables, approved the final draft.
- Ryan J. Ellis performed the experiments, analysed the data, prepared figures and/or tables, authored or reviewed drafts of the paper, approved the final draft.
- Paul Doughty analysed the data, authored or reviewed drafts of the paper, approved the final draft.
- Craig Moritz conceived and designed the experiments, contributed reagents/materials/analysis tools, authored or reviewed drafts of the paper, approved the final draft.

## Animal Ethics
The following information was supplied relating to ethical approvals (i.e. approving body and any reference numbers):
The Australian National University Animal ethics office approved this research (A2012/14).

## Data Availability
Specimen and measurement data for the specimens and the original DArT SNP coding are available in the Supplemental Files.

## New Species Registration
The following information was supplied regarding the registration of a newly described species:
Publication LSID: urn:lsid:zoobank.org:pub:9EA86EF0-DB81-40ED-9DB9-58DBEF9B59D6.
*Gehyra arnhemica* sp. nov.:
LSID: urn:lsid:zoobank.org:act:33DB9200-D783-4738-9B5A-50A6AF4155C1

*Gehyra gemina* sp. nov.: LSID:urn:lsid:zoobank.org:act:BC6CA755-1F3D-4E03-9A09-8E2F1174CC09.

*Gehyra lauta* sp. nov.: LSID:urn:lsid:zoobank.org:act:02B9D2DC-9499-41E3-BD1E-D0F0C8880F83.

*Gehyra chimera* sp. nov.: LSID:urn:lsid:zoobank.org:act:E2BF5779-651D-4CB4-BF67-B7E286750DB2.

*Gehyra lapistola* sp. nov.: LSID: urn:lsid:zoobank.org:act:37EA721F-35D0-4F22-A5FF-24B63EB2F91C.

*Gehyra calcitectus* sp. nov.: LSID: urn:lsid:zoobank.org:act:52919622-D6CD-4023-80B5-3ABEF515DB29.

## Supplemental Information

Supplemental information for this article can be found online at http://dx.doi.org/10.7717/peerj.7971#supplemental-information.

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
