# Peer review of "Crypsis and convergence: integrative taxonomic revision of the Gehyra australis group (Squamata: Gekkonidae) from northern Australia"

_PeerJ, doi:10.7717/peerj.7971_

## Round 0.1 · original submission · Minor Revisions

The reviewers were enthusiastic about the study, but two of the reviewers identified a few minor issues that have to be addressed before the paper can be accepted.

Reviewer 1 ·

Basic reporting

The manuscript is well written, follows up on previously published work, and sets up the argument as to why these new species should be recognized (even if it might be somewhat controversial to do so). While I may not entirely agree with argument supporting recognizing these as species, I am perfectly happy to let the community as a whole decide whether they choose to accept these taxonomic decisions.

Experimental design

Well designed. I recognize no flaws.

Validity of the findings

The authors build a solid argument for supporting their conclusions. The taxonomic decisions are as thorough and well supported as they can be (given the lack of morphological differentiation). This will provide a solid basis for future work should others wish to critique their taxonomic decisions.

Additional comments

Excellent article. Well done to all involved.

·

Basic reporting

The paper is well structured and mostly clear in its exposition. In a few places (noted below) there are some redundancies and minor grammatical lapses or text flaws, but these will be easily remedied.

Experimental design

This is a well-conceived piece of work, building explicitly on recent preliminary studies of the species-level diversity of a group of lizards that have been notoriously difficult to pin down. The MS reports on the introduction of new methods (new both for this group and for such studies in general) that take earlier, tentative conclusions regarding the possibility of cryptic species and thoroughly validate them with much new analysis and a formal taxonomic treatment. The Discussion provides a useful review of the modern integrative approach to studies of cryptic speciation. It is a very valuable addition to the literature, a how-to for workers trying to unravel morphologically confusing species complexes.

Validity of the findings

I found no systemic failures in the MS. The work provides all the necessary background information to assure the it is based on solid data, carefully and appropriately interpreted.

Additional comments

Two general comments:

1. Use of the word “complex” implies relatedness of the complex members, but in this MS, the morphologically aberrant G. chimera is excluded from the G. koira complex, even though this exclusion renders the G. koira complex paraphyletic. Far better if it were included, as this would preserve the idea of its close relationship with the other G. koira complex members, and reinforce its convergent similarity to the G. australis complex. By all means mention that its appearance is deceptive, but do not set it apart from its close genetic relatives.

2. A distant background to this study was King’s initial work on karyotypic variation in Gehyra – how many of these cryptic species were sampled in King’s surveys? Did Moritz get any additional sampling? As this paper mentions chromosomal evolution in the group, it would be interesting to note in the discussion or in the descriptions, if known, cases where the new species share the same karyotype, and if any showed karyotypic differences.

Reviewer 3 ·

Basic reporting

I have had the pleasure of reading the manuscript ‘Convergence and crypsis: Integrative taxonomic revision of the Gehyra australis group (Squamata: Gekkonidae) from northern Australia’ by Oliver et al. This article is sensibly organized and methodologically sound. I do not hesitate to accept this MS with minor revision. There are some minor issues with organization and grammar.

Experimental design

I would also urge authors to consider running and additional analysis to test for actual migration like g-Phocs and/or EMS. I say this because there is extensive talk about reproductive isolation based of a population assignment method. Some explicit test of the magnitude of migration is needed here. Other than that, I think that the methods are fine.

Validity of the findings

Given the quantity and quality of the data collected here, I see no issue with these descriptions.

Additional comments

Line 24: ‘have been revealing’ to ‘have revealed’
Line 38: reword sentence
Line 73: Describe what you mean by ‘derive features’
Line 91: ‘Here’ to ‘Here,’
Line 106: reword this sentence
Line 131: Do you mean genotyped?
Line 174: ‘Contaminations’ to ‘contaminants’
Line 277: Describe differentiation in more detail
Line 286: Describe ancestry proportions from conStruct in more detail
Line 319: You need more detail here. What explained most of the variance between groups?
Line 805: ‘yield’ to ‘yielded’
Line 1294: dataset?
Line 1332: surrounding areas?
Line 1336: ‘However perhaps’ to ‘However,’
Line 1345: switch instances to of ‘low dispersal’ to ‘low vagility’
Line 1350: Clarify sentence
Line 1352: ‘underestimate substantially’ to ‘substantially underestimates’
Figure 2: This figure legend needs more detail. Like that the bottom right one is a biome map, etc. Also, I don’t see an arrow.

Annotated reviews are not available for download in order to protect the identity of reviewers who chose to remain anonymous.

---

## Round 0.2 · accepted · Accept

I believe all of the (minor) issues were properly addressed in the new version of the manuscript.